# Enabling Adaptive Agent Training in Open-Ended Simulators by Targeting Diversity

**Robby Costales**[*]    **Stefanos Nikolaidis**

Department of Computer Science
University of Southern California

## Abstract

The wider application of end-to-end learning methods to embodied decision-making domains remains bottlenecked by their reliance on a superabundance of training data representative of the target domain. Meta-reinforcement learning (meta-RL) approaches abandon the aim of zero-shot *generalization*—the goal of standard reinforcement learning (RL)—in favor of few-shot *adaptation*, and thus hold promise for bridging larger generalization gaps. While learning this meta-level adaptive behavior still requires substantial data, efficient environment simulators approaching real-world complexity are growing in prevalence. Even so, hand-designing sufficiently diverse and numerous simulated training tasks for these complex domains is prohibitively labor-intensive. Domain randomization (DR) and procedural generation (PG), offered as solutions to this problem, require simulators to possess carefully-defined parameters which directly translate to meaningful task diversity—a similarly prohibitive assumption. In this work, we present **DIVA**, an evolutionary approach for generating diverse training tasks in such complex, open-ended simulators. Like unsupervised environment design (UED) methods, DIVA can be applied to arbitrary parameterizations, but can additionally incorporate realistically-available domain knowledge—thus inheriting the *flexibility* and *generality* of UED, and the supervised *structure* embedded in well-designed simulators exploited by DR and PG. Our empirical results showcase DIVA's unique ability to overcome complex parameterizations and successfully train adaptive agent behavior, far outperforming competitive baselines from prior literature. These findings highlight the potential of such *semi-supervised environment design* (SSED) approaches, of which DIVA is the first humble constituent, to enable training in realistic simulated domains, and produce more robust and capable adaptive agents. Our code is available at https://github.com/robbycostales/diva.

## 1 Introduction

Despite the broadening application of reinforcement learning (RL) methods to real-world problems [1, 2], generalization to *new scenarios*—ones not explicitly supported by the training set—remains a fundamental challenge [3]. Meta-reinforcement learning (meta-RL), an extension of the RL framework, is formulated specifically for training *adaptive agents*, and is thus well-suited for overcoming these generalization gaps [4]. One recent work has demonstrated that meta-RL agents can be trained at scale to achieve adaptation capabilities on par with human subjects [5]. However, learning this human-like adaptive behavior naturally requires a large amount of data representative of the downstream (or *target*) distribution. For task distributions approaching real-world complexity—precisely the ones of interest—designing each scenario by hand is prohibitively expensive.

---

[*]Correspondence to `rscostal@usc.edu`.

38th Conference on Neural Information Processing Systems (NeurIPS 2024).

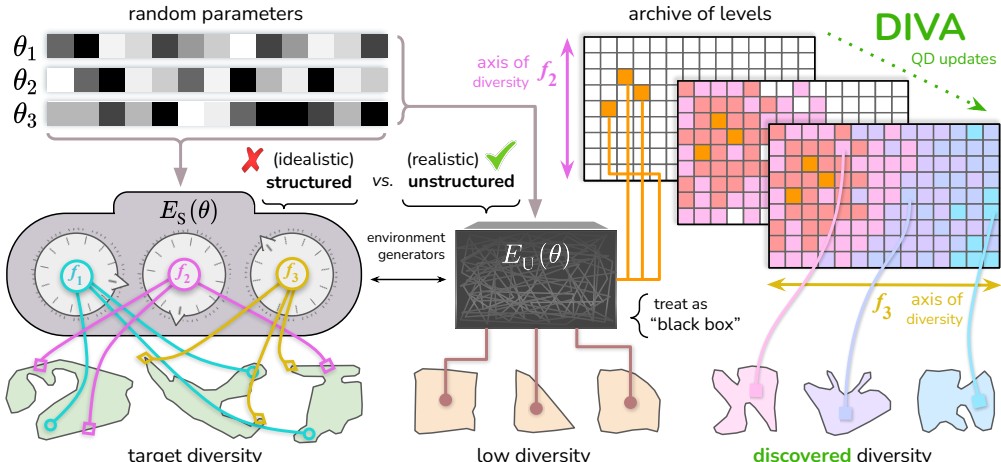

Figure 1: Highly *structured* environment simulators assume access to parameterizations $E_S(\boldsymbol{\theta})$ for which random seeds $\boldsymbol{\theta}_i$ *directly* produce meaningfully diverse features (e.g. RACING tracks with challenging turns). Open-ended environments with flexible, *unstructured* parameterizations $E_U(\boldsymbol{\theta})$—though enabling more complex *emergent* features—lack direct control over high-level features of interest. We introduce **DIVA**, an approach that effectively creates a more workable parameterization $E_{QD}(\boldsymbol{\theta})$ by evolving levels beyond the minimally diverse population from $E_U(\boldsymbol{\theta})$. By training on these discovered levels, DIVA enables superior performance on downstream tasks.

Prior works have explored the use of domain randomization (DR) and procedural generation (PG) techniques to produce diverse training data for learning agents [6]. Despite eliminating the need for hand-designing each task individually, human labor is still required to carefully design an environment generator that can produce diverse, high-quality tasks. As environments become more complex and open-ended, the ability to hand-design such a robust generator becomes increasingly infeasible. Some methods, like PLR [7], attempt to ameliorate this limitation by learning a curriculum over the generated levels, but these works still operate under the assumption that the generator produces meaningfully diverse levels with a high probability.

Unsupervised environment design (UED) [8] are a broad class of appproaches which use performance-based metrics to adaptively form a curriculum of training levels. ACCEL [9], a state-of-the-art UED method, uses an evolutionary process to discover more interesting regions of the simulator's parameter space (i.e. appropriately challenging tasks) than can be found by random sampling. While UED approaches are designed to be generally applicable and require little domain knowledge, they implicitly require a very constrained environment generator—one in which all axes of difficulty correspond to meaningful learning potential for the downstream distribution. Moreover, when faced with complex open-ended environments with arbitrary parameterizations, even ACCEL is not able to efficiently explore the solution space, as it is still bottlenecked by the speed of agent evaluations.

In this work, we introduce **DIVA**, an approach for generating diverse training tasks in open-ended simulators to train adaptive agents. By using quality diversity (QD) optimization to efficiently explore the solution space, DIVA bypasses the problem of needing to evaluate agents on all generated levels. QD also enables fine-grained control over the axes of diversity to be captured in the training tasks, allowing the flexible integration of task-related prior knowledge from both domain experts and learning approaches. We demonstrate that DIVA, with limited supervision in the form of feature samples from the target distribution, significantly outperforms state of the art UED approaches—despite the UED approaches being provided with significantly more interactions. We further show that UED techniques can be integrated into DIVA. Preliminary results with this combination (which we call DIVA+) are promising, and suggest an exciting avenue for future work.

## 2 Preliminaries

**Meta-reinforcement learning.** We use the meta-reinforcement learning (meta-RL) framework to train adaptive agents, which involves learning an adaptive policy $\pi_\phi$ over a distribution of tasks $\mathcal{T}$.

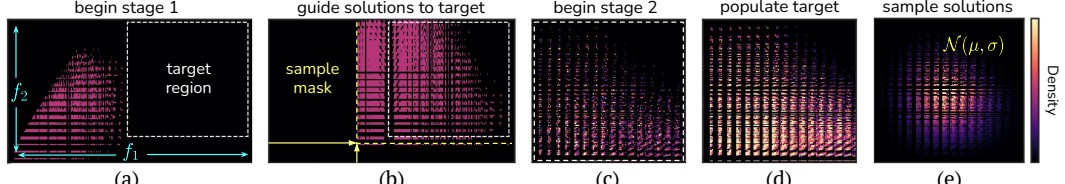

Figure 2: **DIVA archive updates** on ALCHEMY. The *first stage* (a) begins with bounds that encapsulate initial solutions, and the target region. As the first stage progresses (b), and QD discovers more of the solution space, the sampling region for the emitters gradually shrinks towards the target region. The *second stage* begins by redefining the archive bounds to be the target region and including some extra feature dimensions (c). QD fills out just the target region now (d), using sample weights from the target-derived prior (e), the same distribution used to sample levels during meta-training.

Each $\mathcal{M}_i \in \mathcal{T}$ is a Markov decision process (MDP) defined by a tuple $\langle \mathcal{S}, \mathcal{A}, P, R, \gamma, T \rangle$, where $\mathcal{S}$ is the set of states, $\mathcal{A}$ is the set of actions, $P(s_{t+1}|s_t, a_t)$ is the transition distribution between states given the current state and action, $R(s_t, a_t)$ is the reward function, $\gamma \in [0, 1]$ is the discount factor, and $T$ is the horizon. Meta-training involves sampling tasks $\mathcal{M}_i \sim \mathcal{T}$, collecting trajectories $\mathcal{D} = \{\tau^h\}_{h=0}^H$—where $H$ is the number of *episodes* in each *trial* $\tau$ pertaining to the $\mathcal{M}_i$—and optimizing policy parameters $\phi$ to maximize the expected discounted returns across all episodes.

VariBAD [10] is a context variable-based meta-RL approach which belongs to the wider class of RNN-based methods [11, 12]. While prior methods [13, 14] also use context variables to assist in task adaptation, VariBAD uniquely learns within a belief-augmented MDP (BAMDP) $\langle \mathcal{S}, \mathcal{A}, \mathcal{Z}, P, R, \gamma, T \rangle$ where the context variables $z \in \mathcal{Z}$ encodes the agent's uncertainty about the task, promoting Bayesian exploration. VariBAD utilizes an RNN-based variational autoencoder (VAE) to model a posterior belief over possible tasks given the full agent trajectory, permitting efficient updates to prior beliefs.

**Quality diversity.** For a given problem, quality diversity (QD) optimization framework aims to generate a set of diverse, high-quality solutions. Formally, a problem instance of QD [15] specifies an objective function $J : \mathbb{R}^n \to \mathbb{R}$ and $k$ features $f_i : \mathbb{R}^n \to \mathbb{R}$. Let $S = \boldsymbol{f}(\mathbb{R}^n)$ be the feature space formed by the range of $f$, where $\boldsymbol{f} : \mathbb{R}^n \to \mathbb{R}^k$ is the joint feature vector. For each $\boldsymbol{s} \in S$, the QD objective is to find a solution $\boldsymbol{\theta} \in \mathbb{R}^n$ where $\boldsymbol{f}(\boldsymbol{\theta}) = \boldsymbol{s}$ and $J(\boldsymbol{\theta})$ is maximized. Since $\mathbb{R}^k$ is continuous, an algorithm solving the QD problem definition above would require unbounded memory to store all solutions. QD algorithms in the MAP-Elites [16] family therefore discretize $S$ via a tessellation method, where $\mathcal{G}$ is a tessellation of the continuous feature space $S$ into $N_{\mathcal{G}}$ cells. In employing a MAP-Elites algorithm, we relax the QD objective to find a set of solutions $\boldsymbol{\theta}_i, i \in \{1, \ldots, N_{\mathcal{G}}\}$, such that each $\boldsymbol{\theta}_i$ occupies one unique cell in $\mathcal{G}$. We call the occupants $\boldsymbol{\theta}_i$ of all $M$ cells, each with its own position $\boldsymbol{f}(\boldsymbol{\theta}_i)$ and objective value $J(\boldsymbol{\theta}_i)$, the *archive* of solutions.

## 3 Problem setting

One assumption underlying UED methods is that random parameters—or parameter *perturbations* for ACCEL —produce meaningfully different levels to justify the expense of computing objectives on *each* newly generated level. However, when the genotype is not *well-behaved*—when meaningful diversity is rarely generated through random sampling or mutations—these algorithms waste significant time evaluating redundant levels. In our work, we discard the assumption of well-behaved genotypes in favor of making fewer, more realistic assumptions about complex environment generators. There are several assumptions we make about the simulated environments DIVA has access to.

**Genotypes.** We assume access to an unstructured environment parameterization function $E_U(\boldsymbol{\theta})$, where each $\boldsymbol{\theta}$ is a *genotype* (corresponding to the QD solutions $\boldsymbol{\theta}_i$) describing parameters to be fed into the environment generator. QD algorithms can support both continuous and discrete genotype spaces, and in this work we evaluate on domains with both kinds. Crucially, we make no assumption of the *quality* of the training tasks produced by this random generator. We only assume that (1) there is some nonzero (and for practical purposes, nontrivial) probability that this generator will produce a *valid* level for training—one in which success is possible and positive rewards are in reach; and (2) that it is computationally feasible to discover meaningful feature diversity through an intelligent search over the parameter space—an assumption implicit in all QD applications.

**Features.**  We assume access to a pre-defined set of features, $S = \boldsymbol{f}(\mathbb{R}^n)$, that capture axes of diversity which accurately characterize the diversity to be expected within the downstream task distribution. It is also possible to learn or select good environment features from a sample of tasks from the downstream distribution, which we discuss in Section 7. For the sake of simplicity, we use a *grid archive* as our tessellation $\mathcal{G}$, where the $k$ dimensions of the discrete archive correspond to the defined features. The number of bins for each feature is a hyperparameter, and can be learned or adapted over the course of training. We generally find it to be helpful to use moderately high resolutions to ease the search, since smaller leaps in feature-level diversity are required to uncover new cells. By default, we use 100 sample feature values across all domains, but demonstrate in ablation studies that that significantly fewer may be used (see Appendix C).

## 4   DIVA

DIVA assumes access to a small set of feature samples representative of the target domain. It does not, however, require access to the underlying levels themselves. This is a key distinction, as the former is a significantly weaker assumption. Consider the problem of training in-home assistive robots in simulation with the objective of adapting to real-world houses. It is more likely we have access to publicly available data describing typical houses—dimensions, stylistic features, etc.—than we have access to corresponding simulator parameters which produce those exact feature values.

**Feature density estimation.**  DIVA begins by constructing a QD archive with appropriate *bounds* and *resolution*. Given a set of specified *features* $\{f_i\}^k$ and a handful of downstream *feature samples*, we first infer each feature's underlying distribution. These can be approximated with kernel density estimation (KDE), or we can work with certain families of parameterized distributions. For our experiments, we assume each feature is either (independently) normally or uniformly distributed. We use a statistical test[2] to evaluate the fit of each distribution family, and select the best-fitting. Setting the resolution for discrete feature dimensions is straightforward, and depends only on the range. For continuous features, the resolution should enable enough signal for discovering new cells, while avoiding practical issues that arise with too many cells[3]. See Section 5 for domain-specific details.

**Two-stage QD updates.**  Once the feature-specific target distributions are determined, we can use these to set bounds for each archive dimension. A naïve approach would be to set the archive ranges for each feature based on the confidence bounds of the target distribution. However, random samples from $E_{\mathrm{U}}$ may not produce feature values that fall within the target range. We found this to be a major issue in the ALCHEMY domain (see Figure 2), and for some features in RACING. We solve this problem by setting the initial archive bounds to include both randomly generated samples from $E_{\mathrm{U}}$, as well as the full target region. As the updates progress, we gradually update the *sample mask*—which is used to inform the sampling of new solutions—towards the target region. We observe empirically that updating and applying this mask provides an enormous speed-up in guiding solutions towards the target region (see Figure 15). After this first stage, solutions are inserted into a new archive defined by the proper target bounds. See Appendix A for more specifics on these two QD update stages.

**Overview.**  DIVA consists of three stages. Stage 1 (S1) begins by initializing the archive with bounds that include both the downstream feature samples (the *target region*), as well as the initial population generated from $E_U(\theta)$. S1 then proceeds with alternating *QD updates*, to discover new solutions, and *sample mask updates*, to guide the population towards the target region. In Stage 2 (S2), the archive is reinitialized with existing solutions, but is now bounded by the target region. QD updates continue to further diversify the population, now targeting the downstream feature values specifically. The last stage is standard meta-training, where training task parameters are now drawn from $P_{\mathcal{G}}(\boldsymbol{\theta})$, a distribution over the feature space approximated using the downstream feature samples, discretized over the archive cells. See Appendix A for *detailed* pseudocode.

---

**Algorithm 1** DIVA

```
    # Stage 1: discover target region
1:  G ← initialize_archive()
2:  for i in range(N_S1) do
3:      G ← QD_UPDATE(G, J, M, B_QD)
4:      M ← update_sample_mask(M, G)
    # Stage 2: populate target region
5:  G ← update_archive_bounds(G)
6:  for i in range(N_S2) do
7:      G ← QD_UPDATE(G, J, ∅, B_QD)
    # Meta-learn over QD archive
8:  for i in range(N_VB) do
9:      M ← E_U(θ' ∼ P_G(θ))
10:     τ ← perform_rollout(M, π_φ)
11:     D_π ← store_rollout(D_π, τ)
12:     meta_update(π_φ, D_π)
```

---

[2] We use a Kolmogorov–Smirnov test for features with continuous values and Chi-squared for discrete.

[3] Memory is one concern; another is that optimizing *objectives* across *all cells* is slower with more cells.

# 5 Empirical results

**Baselines.** We implement the following baselines to evaluate their relative performance to **DIVA**. **ODS** is the "oracle" agent trained over the downstream environment distribution $E_S(\boldsymbol{\theta})$, used for evaluation. With this baseline, we are benchmarking the upper bound in performance from the perspective of a learning algorithm that has access to the underlying data distribution.[4] **DR** is the meta-learner trained over a task distribution defined by performing domain randomization over the space of valid genotypes, $\boldsymbol{\theta}$, under the training parameterization, $E_U(\boldsymbol{\theta})$. Robust PLR (**PLR$^\perp$**) [17] is the improved and theoretically grounded version of PLR [7], where agents' performance-based PLR objectives are evaluated on each level *before* using them for training. **ACCEL** [9] is the same as PLR$^\perp$ but instead of randomly sampling over the genotype space to generate levels for evaluation, levels are mutated from existing solutions. All baselines use VariBAD [10] as their base meta-learner.

**Experimental setup.** The oracle agent (ODS) is first trained over the each environment's downstream distribution to tune VariBAD's hyperparameters. These environment-specific VariBAD settings are then fixed while hyperparameters for DIVA and the other baselines are tuned. For fairness of comparison—since DIVA is allowed $N_{QD}$ QD update steps to fill its archive before meta-training—we allow each UED approach (PLR$^\perp$ and ACCEL) to use significantly more environment steps for agent evaluations (details discussed below per environment). All empirical results were run with 5 seeds unless otherwise specified, and error bars indicate a 95% confidence region for the metric in question. The QD archive parameters were set per environment, and for ALCHEMY and RACING, relied on some hand-tuning to find the right combinations of features and objectives. We leave it to future work to perform a deeper analysis on what constitutes good archive design, and how to better automate this process.

## 5.1 GRIDNAV

Our first evaluation domain is a modified version of GRIDNAV (Figure 3), originally introduced to motivate and benchmark VariBAD [10]. The agent spawns at the center of the grid at the start of each episode, and receives a slight negative reward ($r = -0.1$) each step until it discovers (inhabits) the goal cell, at which point it also receives a larger positive reward ($r = 1.0$).

**Parameterization.** We parameterize the task space (i.e. the goal location) to reduce the likelihood of generating meaningfully diverse goals. Specifically, each $E_{U_k}$ (or $E_k$) introduces $k$ genes to the solution genotype which together define the final $y$ location. Each gene $j$ can assume the values $\theta_j \in \{-1, 0, 1\}$, and the final $y$ location is determined by summing these values, and performing a floor division to map the bounds back to the original range of the grid. As $k$ increases, $y$ values are increasingly biased towards 0, as shown on the right side of Figure 3. For more details on the GRIDNAV domain, see Appendix B.1.

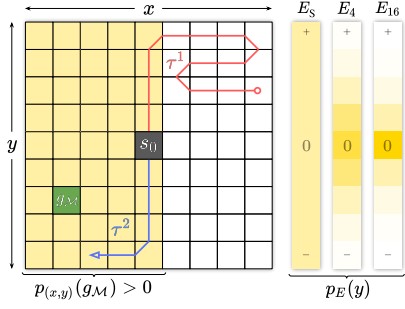

Figure 3: Left: A **GRIDNAV** agent attempting to locate the goal across two episodic rollouts. Right: The marginal probability of sampled goals inhabiting each $y$ for different complexities $k$ of $E_k(\boldsymbol{\theta})$.

**QD updates.** We define the archive features to be the $x$ and $y$ coordinates of the goal location. The objective is set to the current iteration, so that newer solutions are prioritized (additional details in Appendix B.1). DIVA is provided $N_{S2} = 8.0 \times 10^4$ ($N_{S1} = 0$) QD update iterations for filling the archive. To compensate, PLR$^\perp$ and ACCEL are each provided with an additional $9.6 \times 10^6$ environment steps for evaluating PLR scores, which amounts to three times as many total interactions—since all methods are provided $N_E = 4.8 \times 10^6$ interactions for training. If each "reset" call counts as one environment step[5], the UED baselines are effectively granted $2.4\times$ more *additional* step data than what DIVA additionally receives through its QD updates (details in Appendix E.1).

**Results.** From Figure 4a, we see that increasing genotype complexity (i.e. larger $k$) reduces goal diversity for DR—which is expected given the parameterization defined for $E_U$. We can also see that DIVA, as a result of its QD updates, can effectively capture goal diversity, even as complexity

---

[4]Technically, reweighting this distribution (e.g. via PLR) may produce a stronger oracle, but for the purposes of this work, we assume the unaltered downstream distribution can be efficiently trained over, sans curriculum.

[5]In general, rendering the environment (via "reset") is required to compute level features for DIVA.

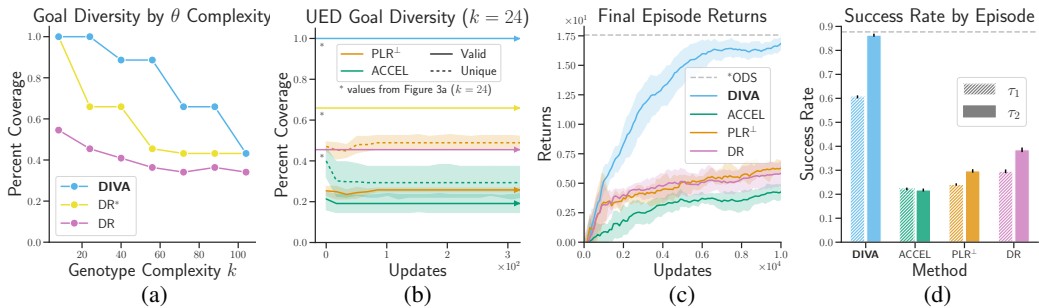

Figure 4: **GRIDNAV analysis and results.** (a) Target region coverage produced by DIVA and DR over different genotype complexities $k$. DR represents the *average* coverage of batches corresponding to the size of the QD archive. DR$^*$ represents the *total number* of unique levels discovered over the course of parameter randomization steps which equal in number to the additional environments PLR$^\perp$ is provided for evaluation. DR$^*$ is thus an upper bound on the diversity that PLR$^\perp$ can capture. 500k iterations (QD or otherwise) are used across all results. (b) The diversity produced by PLR$^\perp$ and ACCEL over the course of training (later updates omitted due to no change in trend). (c) Final episode return curves for DIVA and baselines. (d) Final method success rates across each episode.

increases. When we fix the complexity ($k = 24$) and train over the $E_U$ distribution, we see that the UED approaches are *unable* to incidentally discover and capture diversity over the course of training (Figure 4b). DIVA's explicit focus on capturing meaningful level diversity enables it to significantly outperform these baselines in terms of episodic return (Figure 4c) and success rate (Figure 4d).

## 5.2 ALCHEMY

ALCHEMY [18] is an artificial chemistry environment with a combinatorially complex task distribution. Each task is defined by some *latent chemistry*, which influences the underlying dynamics, as well as agent observations. To successfully maximize returns over the course of a trial, the agent must infer and exploit this latent chemistry. At the start of each episode, the agent is provided a new set of (1-12) *potions* and (1-3) *stones*, where each stone has a *latent state* defined by a specific vertex of a cube, i.e. ($\{0, 1\}, \{0, 1\}, \{0, 1\}$), and each potion has a *latent effect*, or specific manner in which it transforms stone latent states (see Figure 5a). The agent observes only *salient* artifacts of this latent information, and must use interactions to identify the ground-truth mechanics. At each step, the agent can apply any remaining potion to any remaining stone. Each stone's *value* is maximized the closer its latent state is to $(1, 1, 1)$, and rewards are produced when stones are cast into the *cauldron*.

To make training feasible on academic resources, we perform evaluations on the *symbolic* version of ALCHEMY, as opposed to the full Unity-based version. Symbolic ALCHEMY contains the same mechanistic complexity, minus the visuomotor challenges which are irrelevant to this project's aims.

**Parameterization.** $E_S(\boldsymbol{\theta})$ is the downstream distribution containing maximal stone diversity. For training, implement $E_{U_k}$ where $k$ controls the level of difficulty in generating diverse stones. Specifically, we introduce a larger set of coordinating genes $\theta_j \in \{0, 1\}$ that together specify the initial stone latent states, similar to the mechanism we used in GRIDNAV to limit goal diversity. Each stone latent coordinate is specified with $k$ genes, and only when all $k$ *genes* are set to 1 is the *latent coordinate* is set to 1. When *any* of the genes are 0, the latent coordinate is 0. For our experiments we set $k = 8$, and henceforth use $E_U$ to signify $E_{U_8}$.

**QD updates.** We use features LATENTSTATEDIVERSITY and MANHATTANTOOPTIMAL —both of which target stone latent state diversity from different angles. See Appendix B.2 for more specifics on these features and other details surrounding ALCHEMY's archive construction. Like GRIDNAV, the objective is set to bias new solutions. DIVA is provided with $N_{S1} = 8.0 \times 10^4$ and $N_{S2} = 3.0 \times 10^4$ QD update iterations. PLR$^\perp$ and ACCEL are compensated such that they receive 3.5× more *additional* step data than what DIVA receives via QD updates (see Appendix E.1 for details).

**Results.** Our empirical results demonstrate that DIVA is able to generate latent stone states with diversity representative of the target distribution. We see this both quantitatively in Figure 5b, and qualitatively in Figure 6. In Figure 5c, we see this diversity translates to significantly better results

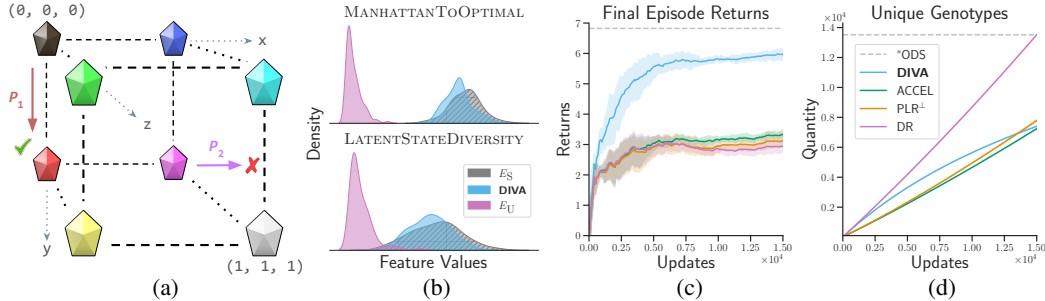

Figure 5: **ALCHEMY environment and results.** (a) A visual representation of ALCHEMY's structured stone latent space. $P_1$ and $P_2$ represent *potions* acting on stones. Only $P_1$ results in a latent state change, because $P_2$ would push the stone outside of the valid latent lattice. (b) Marginal feature distributions for $E_S$ (the structured target distribution), DIVA, and $E_U$ (the unstructured distribution used directly for DR, and to initialize DIVA's archive). (c) Final episode return curves for DIVA and baselines. (d) Number of unique genotypes used by each method over the course of meta-training.

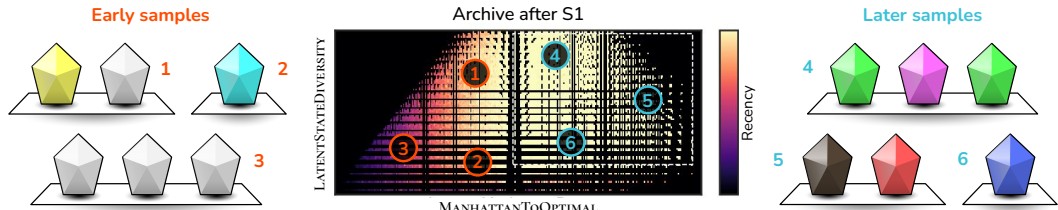

Figure 6: **ALCHEMY level diversity.** Early on in DIVA's QD updates (left), the levels in the archive do not posses much latent stone diversity—all are close to $(1, 1, 1)$. As samples begin populating the target region in later QD updates (right), we see stone diversity is significantly increased.

on $E_S$ over baselines. Despite generating roughly as many unique *genotypes* as DIVA (Figure 5d), PLR$^\perp$ and ACCEL are unable to generate training stone sets of significant *phenotypical* diversity to enable success on the downstream distribution.

### 5.3 RACING

Lastly, we evaluate DIVA on the RACING domain introduced by [17]. In this environment, the agent controls a race car via simulated steering and gas pedal mechanisms, and is rewarded for efficiently completing the track, $\mathcal{M}_i \in \mathcal{T}$. We adapt this RL environment to the meta-RL setting by lowering the resolution of the observation space significantly. By increasing the challenge of perception, even competent agents benefit from multiple episodes to better understand the underlying track. For all of our experiments, we use $H = 2$ episodes per trial, and a flattened $15 \times 15$ pixel observation space.

**Setup.** We use three different parameterizations in our experiments: (1) $E_S(\boldsymbol{\theta})$ is the downstream distribution we use for evaluating all methods, training ODS, and setting archive bounds for DIVA. Parameters $\boldsymbol{\theta}$ are used to seed the random generation of *control points* which in turn parameterize a sequence of Bézier curves designed to smoothly transition between the control locations. Track diversity is further enforced by rejecting levels with control points that possess a standard deviation below a certain threshold. (2) $E_{U_k}(\boldsymbol{\theta})$ is a reparameterization of $E_S(\boldsymbol{\theta})$ that makes track diversity harder to generate, with the difficulty proportional to the value of $k \in \mathbb{N}$. For our experiments, we use $k = 32$ (which we will denote simply as $E_U(\boldsymbol{\theta})$), which roughly means that meaningful diversity is $32\times$ less likely to randomly occur than when $k = 1$ (which is equivalent to $E_S(\boldsymbol{\theta})$). This is achieved by defining a small region in the center, 32 (or $k$, in general) times smaller than the track boundaries, where all points outside the region are projected onto the unit square, and scaled to the track size. (3) $E_{F1}(\boldsymbol{\theta})$ uses $\boldsymbol{\theta}$ as an RNG seed to select between a set of 20 hand-crafted levels official Formula-1 tracks [17], and is used to benchmark DIVA's zero-shot generalization to a new target distribution.

**QD updates.** We define features TOTALANGLECHANGES (TAC) and CENTEROFMASSX (CX) for the archive dimensions. Levels from $E_U$ lack curvature (see Figure 8) so TAC, which is defined as the sum of angle changes between track segments, is useful for directly targeting this desired curvature.

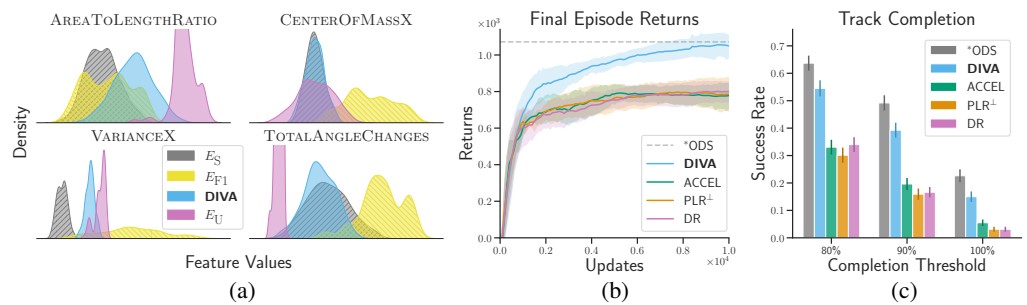

(a)             (b)             (c)

Figure 7: **RACING features and main results.** Left: Marginal feature distributions for $E_{\mathrm{S}}$ (target distribution), $E_{\mathrm{F1}}$ (human-designed F1 tracks), DIVA, and $E_{\mathrm{U}}$ (the unstructured distribution used for DR, the original levels that DIVA evolves)—cropped for readability. Center: Final episode return curves for DIVA and baselines on $E_{\mathrm{S}}$. Right: Track completion rates by method, evaluated on $E_{\mathrm{S}}$.

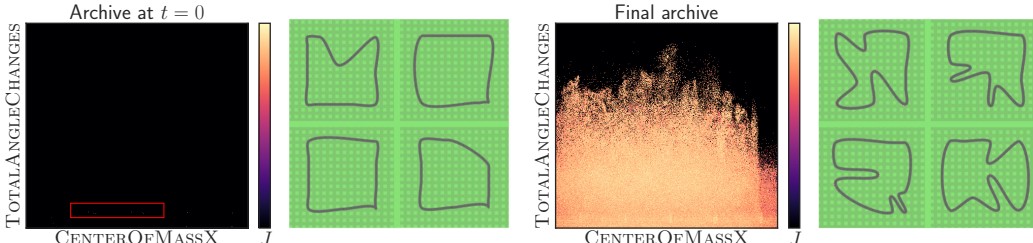

Figure 8: **RACING level diversity.** We see that random $E_{\mathrm{U}}$ levels, used by DR, and which form the initial population of DIVA, are unable to produce qualitatively diverse tracks (left). After the two-stage QD-updates, DIVA is able to produce tracks of high qualitative diversity (right).

CX, or the average location of the segments, targets diversity in the location of these high-density (high-curvature) regions. We compute an *alignment* objective over features CENTEROFMASSY and VARIANCEY to further target downstream diversity. See Appendix B.3 for more details relevant to the archive construction process for RACING. DIVA is provided with $2.5 \times 10^5$ initial QD updates on RACING. PLR$^{\perp}$ and ACCEL are compensated with $4.0\times$ more *additional* step data than what DIVA receives through QD updates (see Appendix E.1 for more details).

**Main results.** Results are shown in Figure 7. DIVA outperforms all baselines, including the UED approaches, which have access to three times as many environment interactions. From Figure 8, we see that final DIVA levels contain significantly more diversity than randomization over $E_{\mathrm{U}}$.

**Transfer to F1 tracks.** Next, we evaluate the ability of these trained policies to zero-shot transfer to human-designed F1 levels [17], $E_{\mathrm{F1}}$. Though qualitative differences are apparent (see Figure 9), from Figure 7a we can additionally see how these levels differ quantitatively. Even though DIVA uses feature samples from $E_{\mathrm{S}}$ to define its archive, we see from the results in Figure 9 that DIVA is not only able to complete many of these tracks, but is also able to significantly outperform ODS. This result may seem unlikely, given that DIVA bases its axes of diversity on $E_{\mathrm{S}}$. One possible explanation is that while DIVA successfully matches its TOTALANGLECHANGES distribution to $E_{\mathrm{S}}$ (see Figure 7), because it is less likely for all 12 control points to be mutated to the diversity-enabling region than just a few control points with sharp angles, DIVA "opts" for the latter, and thus produces fewer, *sharper* angles, which is evidently useful for transferring to (*these*) human-designed tracks. This hypothesis matches what we see qualitatively from the DIVA-produced levels in Figure 8.

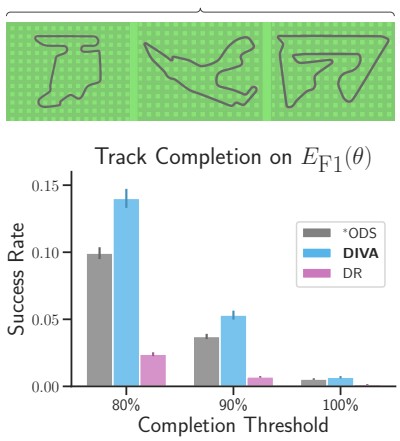

Figure 9: Sample F1 levels (top), and track completion rates by methods targeting $E_{\mathrm{S}}$, evaluated on $E_{\mathrm{F1}}$ (bottom).

**Combining DIVA and UED.** While PLR$^\perp$ and ACCEL struggle on our evaluation domains, they still have utility of their own, which we hypothesize may be compatible with DIVA's. As a preliminary experiment to evaluate the potential of such a combination, we introduce **DIVA+**, which still uses DIVA to generate diverse training samples via QD, but additionally uses PLR$^\perp$ to define

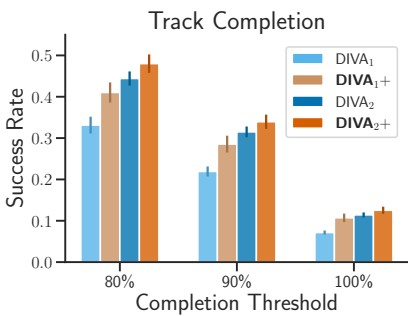

Figure 10: **DIVA+ results** compared to DIVA, for (1) misspecified, and (2) well-specified archives, evaluated on $E_S$.

a new distribution over these levels based on approximate learning potential. Instead of randomly sampling levels from $E_U$, the PLR$^\perp$ evaluation mechanism samples levels from the DIVA-induced distribution over the archive. We perform experiments on two different archives generated by DIVA: (1) an archive that is slightly misspecified (see Appendix B.3 for details), and (2) the archive used in our main results. From Figure 10, we see that while performance does not significantly improve for (2), the combination of DIVA and PLR$^\perp$ is able to significantly improve performance on (1), and even statistically match the original DIVA results. These results highlight the potential of such hybrid (QD+UED) semi-supervised environment design (SSED) approaches, a promising area for future work.

## 6 Related work

**Meta-reinforcement learning.** Meta-reinforcement learning methods range from gradient-based approaches (e.g. MAML) [19], RNN context-based approaches [12, 11] (e.g. RL$^2$), and the slew of emerging works utilizing transformers [20, 5, 21]. We use VariBAD [10], a state-of-the-art context variable-based approach that extends RL$^2$ by using variational inference to incorporate task uncertainty into its beliefs. HyperX [22], an extension that uses reward-bonuses, was not found to improve performance on our domains. In each of these works, the training distribution is given; none address the problem of generating diverse training scenarios in absence of such a distribution.

**Procedural environment generation.** Procedural (content) generation (PCG / PG) [6] is a vast field. Many RL and meta-RL domains themselves have PG baked-in (e.g. ProcGen [23], Meta-World, [24], Alchemy [18], and XLand [5]). Each of these works rely on human engineering to produce levels with meaningfully diverse features. A related stream of works apply scenario generation to robotics—some works essentially perform PCG [25, 26], while others integrate more involved search mechanics [27, 28, 29, 30]. One prior work [31] defines a formal but generic parameterization for applying PG to generate meta-RL tasks. It is yet to be shown, however, if such an approach can scale to domains with vastly different dynamics, and greater complexity.

**Unsupervised environment design.** UED approaches—which use behavioral metrics to automatically define and adapt a curriculum of suitable tasks for agent training—form the frontier of research on open-endedness. The recent stream of open-ended agent/environment co-evolution works (e.g. [32, 33, 34]) was kickstarted by the POET [35, 36] algorithm. The "UED" term itself originated in PAIRED [8], which uses the performance of an "antagonist" agent to define the curriculum for the main (protagonist) agent. PLR [7] introduces an approach for weighting training levels based on *learning potential*, using various proxy metrics to capture this high-level concept. [17] introduces PLR$^\perp$, which only trains on levels that have been previously evaluated, and thus enabling certain theoretical robustness guarantees. AdA [5] uses PLR as a cornerstone of their approach for generating diverse training levels for adaptive agents in a complex, open-ended task space. ACCEL [9] borrows PLR$^\perp$'s scoring procedure, but the best-performing solutions are instead mutated, so the buffer not only collects and prioritizes levels of higher learning potential, but *evolves* them. We use ACCEL as our main baseline because it has demonstrated state-of-the art results on relevant domains, and like DIVA, evolves a population of levels. The main algorithmic differences between ACCEL and DIVA are that ACCEL (1) performs additional evaluation rollouts to produce scores during training and (2) uses a 1-d buffer instead of DIVA's multi-dimensional archive. PLR$^\perp$ serves as a secondary baseline in this work; its non-evolutionary nature makes it a useful comparison to DR.

**Scenario generation via QD.** A number of recent works apply QD to simulated environments in order to generate diverse scenarios, with distinct aims. Some works, like DSAGE [37], uses QD to develop diverse levels for the purpose of probing a pretrained agent for interesting behaviors. In another line of work applies QD to human-robot interaction (HRI), and ranges from generating

diverse scenarios [38], to finding failure modes in shared autonomy systems [39] and human-aware planners [40]. DIVA's application of QD inspired by these approaches, as they produce meaningfully diverse environment scenarios, but no prior work exists which applies QD to define a task distribution for agent *training*, much less *adaptive* agent training, or overcoming difficult parameterizations in open-ended environments.

## 7 Discussion

The present work enables adaptive agent training on open-ended environment simulators by integrating the *unconstrained* nature of unsupervised environment design (UED) approaches, with the implicit *supervision* baked into procedural generation (PG) and domain randomization (DR) methods. Unlike PG and DR, which requires domain knowledge to be carefully incorporated into the environment generation process, DIVA is able to *flexibly* incorporate domain knowledge, and can discover *new* levels representative of the downstream distribution. And instead of relying on behavioral metrics to infer a general, ungrounded form of "learning potential", like UED—which becomes increasingly unconstrained and therefore less useful a signal as environments become more complex and open-ended—DIVA is able to *directly* incorporate downstream feature samples to target specific, *meaningful* axes of diversity. With only a handful of downstream feature samples to set the parameters of the QD archive, our experiments (Section 5) demonstrate DIVA's ability to outperform competitive baselines compensated with three times as many environment steps during training.

In its current form, the most obvious limitation of DIVA is that, in addition to assuming access to downstream feature samples, the axes of diversity themselves must be specified. However, we imagine these axes of diversity could be learned automatically from a set of sample levels, or selected from a larger set of candidate features; it may be possible to adapt existing QD works to automate this process in related settings [41]. The present work also lacks a more thorough analysis of what constitutes good archive design. While some amount of heuristic decision-making is unavoidable when applying learning algorithms to specific domains, a promising future direction would be to study how to approach DIVA's archive design from a more algorithmic perspective.

DIVA currently performs QD iterations over the environment parameter space defined by $E_U(\boldsymbol{\theta})$, where each component of the genotype $\boldsymbol{\theta}$ represents some *salient* input parameter to the simulator. Prior works in other domains (e.g. [42]) have demonstrated QD's ability to explore the latent space of generative models. One natural direction for future work would therefore be to apply DIVA to *neural* environment generators (rather than *algorithmic* generators), where $\boldsymbol{\theta}$ would instead correspond to the latent input space of the generative model. If the latent space of these models is more convenient to work with than the raw environment parameters—e.g. due to greater smoothness with respect to meaningful axes of diversity—this may help QD more efficeintly discover samples within the target region. Conversely, DIVA's ability to discover useful regions of the parameter space means these neural environment generators do not need to be "well-behaved", or match a specific target distribution. Since these generative models are also likely to be differentiable, DIVA can additionally incorporate gradient-based QD works (e.g. DQD [15]) to accelerate its search.

Preliminary results with DIVA+ demonstrate the additional potential of combining UED and DIVA approaches. The F1 transfer results (i.e. DIVA outperforming ODS trained directly on $E_S$) further suggest that agents benefit from flexible incorporation of downstream knowledge. In future work, we hope to study more principle integrations of UED and DIVA-like approaches, and to more generally explore this exciting new area of semi-supervised environment design (SSED).

More broadly, now equipped with DIVA, researchers can develop more general-purpose, open-ended simulators, without concerning themselves with constructing convenient, well-behaved parameterizations. Evaluations in this work required constructing our own contrived paramterizations, since domains are rarely released without carefully designed parameterizations. It is no longer necessary to accomodate the assumption made my DR, PG, and UED approaches—that either randomization over the parameter space should produce meaningful diversity, or that all forms of level difficulty ought to correspond to meaningful learning potential. So long as diverse tasks are *possible* to generate, even if sparsely distributed within the paramter space, QD may be used to discover these regions, and exploit them for agent training. Based on the promising empirical results presented in this work, we are hopeful that DIVA will enable future works to tackle even more complicated domains, and assist researchers in designing more capable and behaviorally interesting adaptive agents.

# 8 Reproducibility statement

The source code, along with thorough documentation for reproducing each result in this paper, is publicly available on Github[6]. Even without this code, researchers should be able to fully reproduce the algorithm from the details in the main body, the pseudocode provided in Appendix A, and training details (hyperparameters and hardware information) provided in Appendix E.

# 9 Ethics statement

Like all fundamental technologies, this work has the potential to be misapplied for malicious purposes. The authors do not believe, however, that the methods introduced in this work present a significant or unique risk for misuse or abuse. The authors intend for DIVA to be applied to use-cases that have the best interests of humanity (including concern for the earth and other sentient creatures) at heart.

# 10 Acknowledgements

This work was partially supported by NSF CAREER (#2145077) and the DARPA EMHAT project. We thank Tjanaka et al., the developers of pyribs [43], whose library served as the basis for our QD implementations. We thank Zintgraf et al., the authors of VariBAD [10], whose codebase served as the basis for our meta-RL agent. We thank Jiang et al. and Parker-Holder et al., the authors of PLR [7] and ACCEL [9], respectively, for their implementations which served as the basis for our UED baselines. We specifically thank Minqi Jiang for answering questions related to the PLR codebase in the early stages of development, and Varun Bhatt for helpful discussion at various stages of this work.

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

# Appendix

## A   Algorithmic details

**Algorithm.**   The pseudocode below walks through the entire training process for DIVA in abstract. All of the new components that **DIVA** introduces is written in **green**, and all **DIVA+** modifications are in **blue**. Original **VariBAD** training steps are in **black**, and all inline **comments** are in **orange**.

---

**Algorithm 2** DIVA (detailed)

---

1:  # Initialize VariBAD and QD components
2:  $\pi_\phi \leftarrow$ init_policy();  $f_{\text{enc}}, f_{\text{dec}} \leftarrow$ init_vae()         ▷ Initialize VariBAD components
3:  $\mathcal{D}_\pi, \mathcal{D}_{\text{VAE}} \leftarrow$ init_storage_buffers()         ▷ Initialize VariBAD buffers
4:  $\Theta_0 \leftarrow \{\boldsymbol{\theta}_i\}^{n_0} \sim P(\boldsymbol{\theta})$         ▷ Sample initial solutions from space of valid genotypes
5:  $\boldsymbol{F}_0 = [f(\boldsymbol{\theta}_1), \ldots, f(\boldsymbol{\theta}_{n_0})] \leftarrow$ compute_features($\Theta$)         ▷ Compute env. features
6:  $\boldsymbol{J}_0 = [J(\boldsymbol{\theta}_1), \ldots, J(\boldsymbol{\theta}_{n_0})] \leftarrow$ compute_objectives($\Theta$)         ▷ Compute env. objectives
7:  $\mathcal{G} \leftarrow$ initialize_archive($\boldsymbol{F}_0, \boldsymbol{F}_{\text{S}}$)         ▷ Init. archive to contain both $\boldsymbol{F}_0$ and target features $\boldsymbol{F}_{\text{S}}$
8:  $\mathcal{G} \leftarrow$ insert_solutions($\mathcal{G}, (\Theta_0, \boldsymbol{F}_0, \boldsymbol{J}_0)$)         ▷ Add random solutions to archive

9:  # QD stage 1: *discover* target region
10: **for** i **in** range($N_{\text{S1}}$) **do**
11:     $\mathcal{G} \leftarrow$ QD_UPDATE($\mathcal{G}, J, \boldsymbol{M}, B_{\text{QD}}$)         ▷ Perform QD update (batch size $B_{\text{QD}}$) to populate archive
12:     $\boldsymbol{M} \leftarrow$ update_sample_mask($\boldsymbol{M}, \mathcal{G}$)         ▷ Move mask gradually towards target region

13: # QD stage 2: *populate* target region
14: $\mathcal{G} \leftarrow$ update_archive_bounds($\mathcal{G}$)         ▷ Create final archive (target region) before S2 updates
15: **for** i **in** range($N_{\text{S2}}$) **do**
16:     $\mathcal{G} \leftarrow$ QD_UPDATE($\mathcal{G}, J, \emptyset, B_{\text{QD}}$)         ▷ Perform QD update to populate target region

17: # Meta-learning over QD archive
18: **for** i **in** range($N_{\text{VB}}$) **do**
19:     $\boldsymbol{\theta}' \sim P_{\mathcal{G}}(\Theta)$         ▷ Sample solution $\boldsymbol{\theta}'$ from QD archive using approximated target density from $\boldsymbol{F}_{\text{S}}$
20:     $\mathcal{M} \leftarrow$ generate_environment($\boldsymbol{\theta}'$)         ▷ Generate new training environment from solution $\boldsymbol{\theta}'$
21:     # Produce meta-RL rollouts
22:     $\tau \leftarrow$ perform_policy_rollout($\mathcal{M}, \pi_\phi$)
23:     $\mathcal{D}_\pi, \mathcal{D}_{\text{VAE}} \leftarrow$ add_to_buffers($\mathcal{D}_\pi, \mathcal{D}_{\text{VAE}}, \tau$)

24:     # Update VAE and policy
25:     $f_{\text{enc}}, f_{\text{dec}} \leftarrow$ varibad_vae_update($f_{\text{enc}}, f_{\text{dec}}, \mathcal{D}_{\text{VAE}}$)
26:     **if** after_vae_pretraining() **then**
27:         varibad_policy_update($\pi_\phi, \mathcal{D}_\pi$)

28:     # Perform DIVA+ QD updates
29:     **if** DIVA+ **and** $(i\ \%\ $qd_update_interval$\ =\ 0)$ **then**
30:         **for** qd_updates_per_iter **do**
31:             $\mathcal{G} \leftarrow$ QD_UPDATE($\mathcal{G}, J_{\text{PLR}\perp}, \emptyset, B_{\text{QD}}$)         ▷ Perform QD update *with PLR objective*

---

**Algorithm 3** QD update

---

1:  # Perform a single QD update on archive $\mathcal{G}$ with batch size $B$.
2:  **function** QD_UPDATE($\mathcal{G}, J, \boldsymbol{M}, B$)
3:     $\tilde{\Theta}^{B \times n} = [\tilde{\boldsymbol{\theta}}_1, \ldots, \tilde{\boldsymbol{\theta}}_B] \leftarrow$ sample_from_emitters($\mathcal{G}, \boldsymbol{M}, B$)         ▷ Get mutated batch of solutions
4:     $\boldsymbol{F}^{B \times k} = [f(\tilde{\boldsymbol{\theta}}_1), \ldots, f(\tilde{\boldsymbol{\theta}}_B)] \leftarrow$ compute_features($\tilde{\Theta}$)         ▷ Compute env. features
5:     $\boldsymbol{J}^{B \times 1} = [J(\tilde{\boldsymbol{\theta}}_1), \ldots, J(\tilde{\boldsymbol{\theta}}_B)] \leftarrow$ compute_objectives($\tilde{\Theta}$)         ▷ Compute env. objectives
6:     $\mathcal{G}' \leftarrow$ add_solutions($\mathcal{G}, (\tilde{\Theta}, \boldsymbol{F}, \boldsymbol{J})$)         ▷ Add new solutions to archive *if they are elites*
7:     **return** $\mathcal{G}'$

---

**Details on the two-stage QD updates**   Here we provide more details on the process described in Section Section 4. Hyperparameters $N_{\text{S1}}$ and $N_{\text{S2}}$ are set to define the number of QD updates to perform in each stage (see Appendix D). In proportion to how many updates in S1 have elapsed, if the sample mask is enabled, the mask is moved at a linear pace from encapsulating the full S1 archive, to covering only the target region. We also set a hyperparameter, $N_{\text{SM}}$ (see Appendix D), which

specifies the minimum number of solutions which must exist within the mask's new bounds for it to be updated. This is to ensure the mask never outpaces the search process. The mask was only found to be necessary in the ALCHEMY environment. In S1 we sample solutions uniformly from within the mask. In S2, we begin sampling from the discretized target density distribution approximated from the downstream feature samples. Two stages are used for RACING as well, since many initial samples fall outside of the target region, but masking was not found to be necessary. The sample mask has a relatively straightforward implementation for MAP-Elites, which we use for ALCHEMY's discrete genotype (and GRIDNAV, where no mask is required). Since MAP-Elite updates entail performing mutations on solutions directly sampled from the archive, the mask is implemented to only consider solutions that fall within the mask bounds. However, since the CMA-ES-based emitter we use for RACING operates by sampling from a parameterized distribution, instead of sampling from the archive directly, the mask would need to be applied to these parameters instead of the archive.

# B  Domain details

## B.1  GRIDNAV

**GRIDNAV features.**    The following features are defined for the GRIDNAV environment:

Table 1: GRIDNAV features.

| Name | Abbr. | Description |
|------|-------|-------------|
| XPOSITION | XP | *x position of the goal.* |
| YPOSITION | YP | *y position of the goal.* |

## B.2  ALCHEMY

**ALCHEMY features.**    We defined the following features for the ALCHEMY environment:

Table 2: ALCHEMY features.

| Name | Abbr. | Description |
|------|-------|-------------|
| MANHATTANTOOPTIMAL | MTO | *Average Manhattan distance between all stones (across all trials) to the optimal state.* |
| STONETOSTONEDISTANCE | STSD | *Average Euclidean distance between all pairs of stones (across all trials).* |
| GRAPHNUMBOTTLENECKS | GNB | *The number of bottlenecks in the graph topology.* |
| LATENTSTATEDIVERSITY | LSD | *The 'diversity' of the latent stone states (across all trials). Diversity is calculated as the standard deviation of each latent state coordinate across all stones.* |
| PARITYFIRSTPOTION | PFP | *First potion location (first trial, first potion), as a parity measure.* |
| PARITYFIRSTSTONE | PFS | *First stone location (first trial, first stone), as a parity measure.* |
| POTIONEFFECTDIVERSITY | PED | *The 'diversity' of the potion effects (across all trials). Diversity is calculated as the standard deviation of each potion effect coordinate across all potions.* |
| POTIONPERMUTATION | PP | *Potion permutation.* |
| POTIONREFLECTION | PR | *Potion reflection.* |
| STONEREFLECTION | SRE | *Stone reflection.* |
| STONEROTATION | SRO | *Stone rotation.* |
| STONETOSTONEDISTANCEVARIANCE | STSDV | *Variance of the distances between stones (across all trials).* |

Figure 11 contains the feature distributions for the structured and unstructured environment parameterizations on ALCHEMY, computed over 100 feature samples. Figure 14 shows the covariance between feature values for RACING, computed over 100 feature samples.

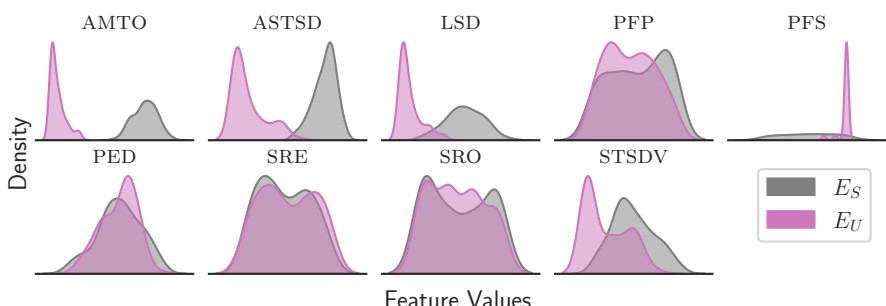

Figure 11: **ALCHEMY all feature distributions.**

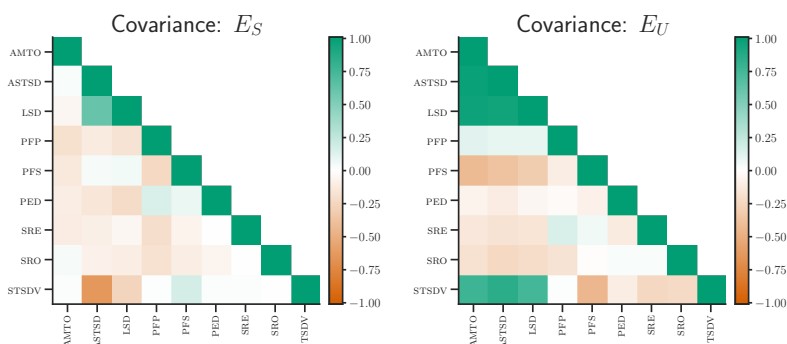

Figure 12: **ALCHEMY measure covariances.**

Archive hyperparameters for ALCHEMY were determined based on some knowledge about the domain, as well as the feature distributions (Figure 13). We noticed a major deviation between $E_U$ and $E_S$ in the feature LATENTSTATEDIVERSITY (LSD), and an even greater on in MANHATTANTOOPTIMAL (MTO). These two constitute the initial dimensions of the archive, and we found the sample mask updates to be crucial to reach and fill the target region (see Figure 15). We use PARITYFIRSTSTONE (PFS) in the second stage to encourage more diversity once the target is reached; it is excluded from the first stage, which is focused on simply reaching the target. The archive for the first stage is of shape $[100, 300, 1]$, corresponding to LSD, MTO, and PFS. The second stage shape is $[150, 150, 5]$. We found this archive to produce diverse enough solutions, evidenced by the number and spread in the target region, so we used this setting to train our DIVA agents. The only objective we found useful for ALCHEMY was a slight bias for newly generated solutions, which we also used for GRIDNAV. We hypothesize this prevents the archive from getting "stuck" with a suboptimal set of solutions, in absense of other objectives.

### B.3 RACING

**RACING features.** See Table 13 for all features defined on RACING. Figure 13 contains the feature distributions for the structured and unstructured environment parameterizations on RACING, computed over 100 feature samples. Figure 14 shows the covariance between feature values for RACING, computed over 100 samples.

Archive hyperparameters for RACING were determined through trial and error, by viewing the samples produced by the archives at the end of the QD updates, as well as the target coverage metrics. After a few iterations, it became clear that Total Angle Change TOTALANGLECHANGES (TAC) was the most useful feature, and so we tried pairing it with a number of others, prioritizing other features with low absolute covariance (see Figure 14).

The best performing archive used by DIVA on RACING uses TAC and CX as its features, and used a measure *alignment* objective over CY and VY. The measure alignment objective rewards solutions for having measure values over the specific measures that are similar to the target distribution. We also

Table 3: RACING features.

| Name | Abbr. | Description |
|---|---|---|
| AREATOLENGTHRATIO | ATLR | *The ratio of enclosed area to curve length.* |
| AVERAGECURVATURE | AC | *The average curvature at midpoints of Beziér segments.* |
| CENTEROFMASSX | CX | *The center of mass x position over the curve.* |
| CENTEROFMASSY | CY | *The center of mass y position over the curve.* |
| CURVEDISTANCESVARIANCE | CDV | *The variability in distances between successive points.* |
| CURVELENGTH | CL | *The total length of the Beziér curve.* |
| ENCLOSEDAREA | EA | *The area enclosed by the Beziér curve.* |
| MEDIANX | MX | *The median x position over the curve.* |
| MEDIANY | MY | *The median y position over the curve.* |
| SIGNIFICANTANGLECHANGES | SAC | *The sum of significant angle changes across the curve.* |
| TOTALANGLECHANGES | TAC | *The total change in angle across the curve.* |
| TOTALCURVATURE | TC | *The total curvature over each segment and sum them up.* |
| VARIANCEX | VX | *The variance of the x positions over the curve.* |
| VARIANCEY | VY | *The variance of the y positions over the curve.* |

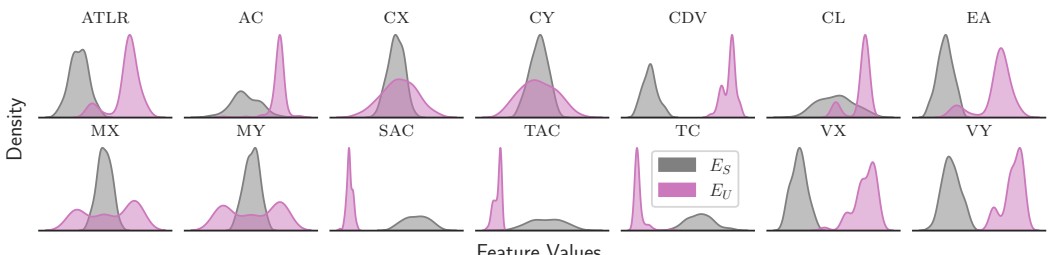

Figure 13: **RACING all feature distributions.**

found that randomly sampling these objective values according to the target distribution provided some additional support in covering the target region efficiently.

The slightly "misspecified" archive was chosen because its solutions generated some diversity, but not as much as the aforementioned one. This archive uses TAC and ATLR as its features, and uses a measure *diversity* objective over just CY. Instead of prioritizing alignment to the target distribution, the diversity objective samples a handful of solutions from the archive, and uses the current solutions deviation from these as its objective.

We use final archive dimensions of $500 \times 500$ for both S1 and S2.

## C Ablation analysis

**Sample mask ablation.** Figure 15 shows the benefit of updating the sample mask bounds during the first archive filling stage on ALCHEMY. Not only does this approach produce significantly more total archive solutions, but more importantly, progress towards filling the target region specifically is accelerated.

## D Hyperparameter sensitivity analysis

**Varying QD mutation rate.** We perform an ablation on the QD mutation rate, which is the probability that a given gene will be mutated (for MAP-Elites). We perform this ablation on ALCHEMY because the its search is the most challenging of the three environments we consider (it is the sole environment that required a longer S1 and the sample mask trick for accelerating to accelerate the search). We see from Figure 16 that ALCHEMY results are not very sensitive to the setting of the mutation rate.

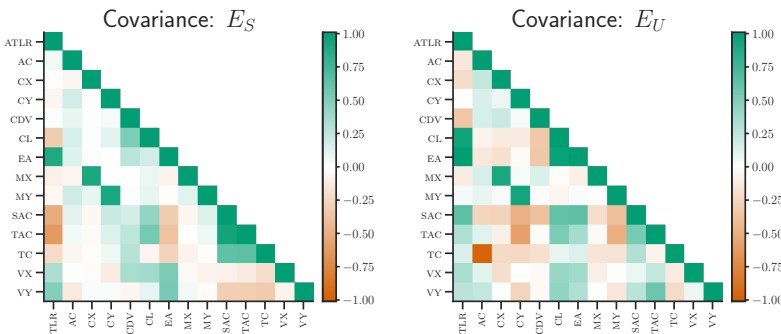

Figure 14: **RACING measure covariances.**

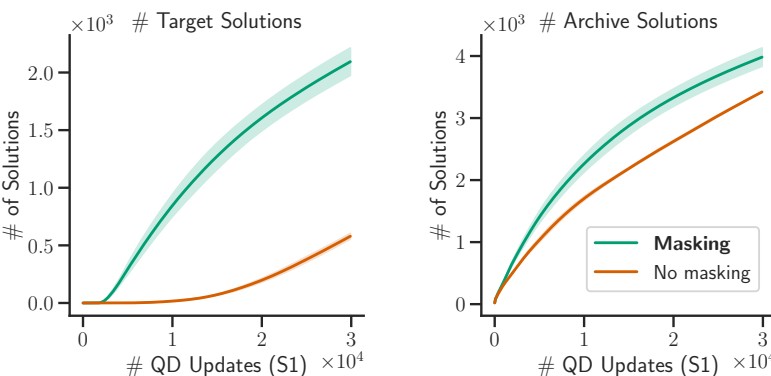

Figure 15: **ALCHEMY sample mask ablation curves.** This specific result is the result of two seeds instead of five, as we found the variance to be very low for this ablation (validated across other parameter settings).

**Varying number of QD updates.** We perform a similar ablation to test how robust DIVA is to the number of QD updates performed. We see from Figure 17 that ALCHEMY results suffer somewhat from fewer updates (e.g. for only 10k in each stage), still significantly outperform baselines in each case. The trend is clear, however: more QD updates produces more solutions, which generally translates to better performance, even if slightly.

**Varying number of downstream samples.** Next we test how robust DIVA is to the number of downstream samples used to compute the target distribution. In Figure 18 we see that, despite the errors increasing with fewer samples, DIVA still significantly outperforms baselines with as few as *five samples*.

# E  Training details

## E.1  DIVA hyperparameters

Table 4 displays the hyperparameters used for DIVA across all domains.

**A note on $N_{\text{TRS}}$ computation** The initial QD population ($n_0$) is implemented such that the first set of QD updates simply generates $n_0$ random levels from $E_U$, before performing the actual mutations (for ME) or intelligent sampling (for ES). Thus, the formula we use for computing $N_{\text{TRS}}$, the *total reset steps* provided to DIVA (see Table 4), which we use to compare the extra steps we provide PLR$^{\perp}$ and ACCEL (discussed in Section 5), does not include $n_0$; it is simply the product of the batch size and the total number of QD iterations.

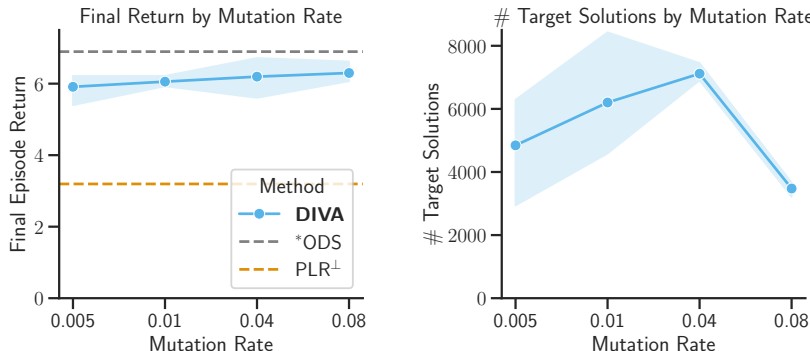

Figure 16: **Effect of varying QD mutation rate in ALCHEMY.** Left: The returns for the final episode by mutation rate, after training on archives produced with each mutation rate. Right: The final number of solutions in the archive after performing QD updates with each mutation rate. This result was produced by running three different seeds for each mutation rate.

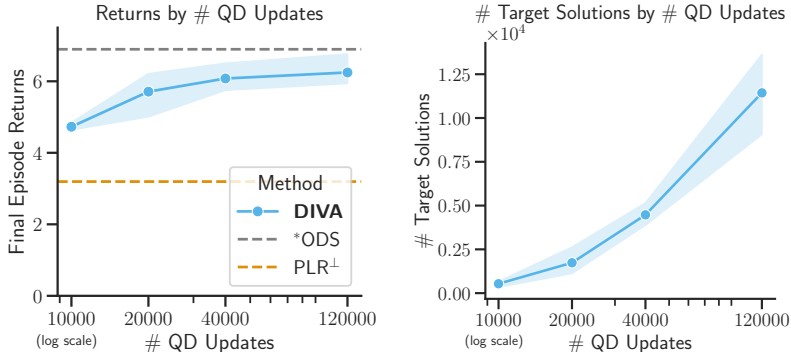

Figure 17: **Effect of varying the number of QD updates in ALCHEMY.** Left: The returns for the final episode by number of QD updates in each stage ($N_{S1} = N_{S2}$). Right: The final number of solutions in the archive after performing each number of QD updates. This result was produced by running three different seeds for each setting.

## E.2    VariBAD hyperparameters

Table 5 displays the hyperparameters used for VariBAD across all domains.

## E.3    Baseline hyperparameters

### E.3.1    PLR$^{\perp}$

Table 6 displays the hyperparameters used for PLR$^{\perp}$ across all domains.

### E.3.2    ACCEL

ACCEL uses the same hyperparameters as PLR$^{\perp}$ (see Table 6), combined with the same evolutionary hyperparameters used for DIVA's QD archive (see Table 4).

## E.4    Computational details

All results were produced on a handful of Titan X or Xp GPUs. Environments were parallelized across multiple CPU cores to accelerate training. While the experiment time varies by method and environment, most experiments take less than a day to run to completion. PLR$^{\perp}$ and ACCEL take the longest, as they required twice as many environment steps as the other methods—on the two latter domains, these methods take well over a day to run to completion.

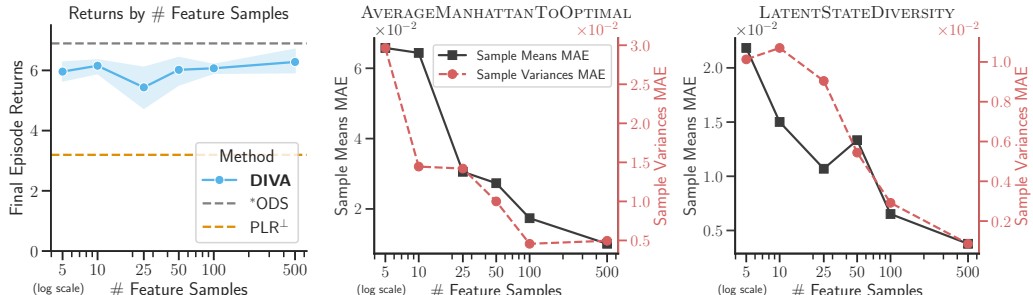

Figure 18: **Effect of varying number of samples in ALCHEMY.** Left: DIVA evaluation returns for the final episode by number of downstream samples, after training on archives produced with by using each number of samples to produce archive bounds and prior. Center/Right: Errors for mean and variance parameters of the normal distribution based on number of samples used for computation; for MANHATTANTOOPTIMAL and LATENTSTATEDIVERSITY. For all plots, five seeds were used for each hyperparameter setting.

Table 4: DIVA hyperparameter settings.

| Name | Description | Value |
|---|---|---|
| $N_{S1}$ | *Number of QD updates in stage* S1 | $0^{\dagger}$ / $80{,}000^{\star}$ / $50{,}000^{\diamond}$ |
| $N_{S2}$ | *Number of QD updates in stage* S2 | $100{,}000^{\dagger}$ / $30{,}000^{\star}$ / $200{,}000^{\diamond}$ |
| $N_{QD}$ | *(Effective) total QD updates:* $N_{S1} + N_{S2}$ | $100{,}000^{\dagger}$ / $110{,}000^{\star}$ / $250{,}000^{\diamond}$ |
| $n_0$ | *Initial population size* | $1{,}000^{\dagger\star}$ / $2{,}000^{\diamond}$ |
| $n_e$ | *Number of QD solution emitters* | $5$ |
| $B_e$ | *Sampling batch size of each QD emitter* | $8^{\dagger\diamond}$ / $5^{\star}$ |
| $B_{QD}$ | *(Effective) QD batch size per update* $(n_e \times B_e)$ | $40^{\dagger\diamond}$ / $25^{\star}$ |
| $N_{TRS}$ | *(Effective) total reset steps* $(N_{QD} \times B_{QD})$ | $4.0 \times 10^6\ ^{\dagger}$ / $2.75 \times 10^6\ ^{\star}$ / $1.0 \times 10^7\ ^{\diamond}$ |
| qd_emitter | *Type of QD emitter* | MAP-Elites (ME)$^{\dagger\star}$ / CMA-ES (CMA) $^{\diamond}$ |
| $p_{ME}$ | *Mutation percentage for ME emitter* | $0.1^{\dagger}$ / $0.02^{\star}$ |
| $\sigma_{ES}$ | *Initial sigma for ES emitter* | $0.1^{\diamond}$ |
| $AB_{SM}$ | *Anneal sample mask bounds during* S1 | False$^{\dagger}$ / True$^{\star\diamond}$ |
| $N_{SM}$ | *Minimum solutions in sample mask* | $40^{\star}$ / $1000^{\diamond}$ |
| $J_{new}$ | *Enable objective for slightly biasing new solutions* | True $^{\dagger\star}$ / False $^{\diamond}$ |
| $J_{MD}$ | *Enable measure diversity objective* | False |
| $J_{MA}$ | *Enable measure alignment objective* | False $^{\dagger\star}$ / True $^{\diamond}$ |
| $J_{Rnd}$ | *Enable randomize objective* | False $^{\dagger\star}$ / True $^{\diamond}$ |
| $N_E$ | *Total meta-training env. steps (ref. from Table 5)* | $4.8 \times 10^6\ ^{\dagger\star}$ / $2.0 \times 10^7\ ^{\diamond}$ |
| $N_{E+}$ | *Additional env. steps provided to UED baselines* | $9.6 \times 10^6\ ^{\dagger\star}$ / $4.0 \times 10^7\ ^{\diamond}$ |
| $N_{E'}$ | *(Effective) total UED env. steps* | $14.4 \times 10^6\ ^{\dagger\star}$ / $6.0 \times 10^7\ ^{\diamond}$ |
| $N_{E+}/N_{TRS}$ | *(Effective) Ratio of add. steps for UED vs DIVA* | $2.4^{\dagger}$ / $3.5^{\star}$ / $4.0^{\diamond}$ |

$^{\dagger}$GRIDNAV, $^{\star}$ALCHEMY, $^{\diamond}$RACING

Table 5: VariBAD hyperparameter settings.

| Name | Description | Value |
| --- | --- | --- |
| $\epsilon_\pi$ | *Optimizer epsilon for policy* | 1e-8 |
| $\gamma$ | *Discount factor for rewards* | 0.99 |
| policy_state_emb_dim | *State embedding dimension for policy* | 64 |
| policy_latent_emb_dim | *Latent embedding dimension for policy* | 64 |
| policy_norm_state | *Normalize state input* | True |
| policy_norm_latent | *Normalize latent input* | True |
| policy_norm_belief | *Normalize belief input* | True |
| policy_norm_rew | *Normalize rewards for policy* | True$^\dagger$ / False$^{\star\diamond}$ |
| policy_layers | *Hidden layers for policy network* | $[128, 128]^{\dagger\star}$ / $[128]^\diamond$ |
| policy_activation | *Activation function for policy* | tanh$^{\dagger\star}$ |
| policy_init_method | *Initialization method for policy* | normc$^{\dagger\star}$ |
| policy_optimizer | *Optimizer for policy* | adam |
| policy_lr | *Learning rate for policy* | 0.0007 |
| policy_init_sd | *Initial standard deviation for policy* | 1.0 |
| policy_val_loss_coef | *Value loss coefficient for policy* | 0.5 |
| policy_entropy_coef | *Entropy coefficient for policy* | 0.01 |
| policy_use_gae | *Use generalized advantage estimation* | True |
| policy_tau | *GAE parameter for policy* | 0.95 |
| policy_max_grad_norm | *Maximum gradient norm for policy* | 0.5 |
| $N_{\text{VB}}$ | *Total number of VariBAD learning updates* | $1.0 \times 10^4$ $^{\dagger\diamond}$ / $1.5 \times 10^4$ $^\star$ |
| policy_num_steps | *Number of environment steps per update* | $60^\dagger$ / $40^\star$ / $500^\diamond$ |
| num_processes | *Number of parallel environments* | $8^{\dagger\star}$ / $4^\diamond$ |
| $N_E$ | *Total env. steps (product of prior three)* | $4.8 \times 10^6$ $^{\dagger\star}$ / $2.0 \times 10^7$ $^\diamond$ |
| ppo_num_epochs | *Number of epochs per PPO update* | $2^{\dagger\star}$ / $8^\diamond$ |
| ppo_num_minibatch | *Number of minibatches for PPO* | 4 |
| ppo_huber_loss | *Use Huber loss for PPO* | True |
| ppo_clip_val_loss | *Use clipped value loss for PPO* | True |
| ppo_clip_param | *Clipping parameter for PPO* | $0.05^{\dagger\star}$ / $0.01^\diamond$ |
| $\alpha_{\text{VAE}}$ | *Learning rate for VAE* | 0.001 |
| $N_{\text{VAE}}$ | *Size of VAE buffer* | 5,000$^\dagger$ / 100,000$^\star$ / 1,000$^\diamond$ |
| $B_{\text{VAE}}$ | *Number of trajectories per VAE update* | $25^{\dagger\star}$ / $10^\diamond$ |
| precollect_len | *Frames to pre-collect before training* | 5,000 |
| num_vae_updates | *Number of VAE update steps per iteration* | 3 |
| pretrain_len | *Number of VAE pre-training updates* | 0 |
| kl_weight | *Weight for KL term* | 0.01 |
| action_emb_size | *Action embedding size for VAE* | 8 |
| state_emb_size | *State embedding size for VAE* | 16 |
| rew_emb_size | *Reward embedding size for VAE* | 16 |
| enc_gru_hidden_size | *GRU hidden size in encoder* | 128 |
| latent_dim | *Latent dimension for VAE* | $10^{\dagger\star}$ / $5^\diamond$ |
| rew_loss_coeff | *Reward loss coefficient* | 1.0 |
| rew_dec_layers | *Layers for reward decoder* | [64, 32] |
| rew_multihead | *Use multihead for reward prediction* | False |
| rew_pred_type | *Reward prediction type* | bernoulli |
| kl_to_gaus_prior | *KL term to Gaussian prior* | False |
| rl_loss_thru_enc | *Backprop RL loss through encoder* | False |
| vae_loss_coef | *VAE loss coefficient* | 1.0 |

$^\dagger$GridNav, $^\star$Alchemy, $^\diamond$Racing

Table 6: PLR hyperparameter settings.

| Name | Description | Value |
|------|-------------|-------|
| $N_{\text{PLR}}$ | *Number of levels to store in our buffer* | $45^{\dagger}$ / $112{,}500^{\star}$ / $8{,}000^{\diamond}$ |
| $f_{\text{PLR}}$ | *Level replay score transform function* | power |
| $\beta_S$ | *Level replay temperature* start | 1.0 |
| $\beta_E$ | *Level replay temperature* end | 1.0 |
| $\textbf{score}(\tau, \pi)$ | *Level replay scoring function* | Positive value loss |
| $\epsilon_{\text{PLR}}$ | *Level replay epsilon for eps-greedy sampling* | 0.05 |
| $p_{\text{replay}}$ | *Probability of sampling replay vs. new level* | 0.5 |
| $\alpha_{\text{PLR}}$ | *Level score EWA smoothing factor* | 0.0 |
| $\rho_C$ | *Staleness coefficient* | 0.7 |
| $f_C$ | *Staleness normalization transform* | power |
| $\beta_C$ | *Staleness normalization temperature* | 1.0 |

$^{\dagger}$GRIDNAV, $^{\star}$ALCHEMY, $^{\diamond}$RACING

