# OpenReview forum: "Enabling Adaptive Agent Training in Open-Ended Simulators by Targeting Diversity"
_NeurIPS.cc/2024/Conference — NeurIPS 2024 poster_

### Official Review · Reviewer_kFbG · 2024-07-12

**Soundness:** 4
**Presentation:** 4
**Contribution:** 4
**Rating:** 8
**Confidence:** 4

**Summary:**

This paper applies quality diversity (QD) optimization (evolutionary method) to the problem of diverse task generation for (meta) reinforcement learning (RL). It argues that QD could be used in settings where an open-ended simulator’s parameterization is unlikely to produce tasks that are diverse in high-level features. The method works by handcrafting a set of high-level task features that are relevant to the learning process of the RL agent, then running QD to collect a set of diverse parameterizations that cover the feature space distribution well. A (meta) RL agent is then trained on tasks sampled from a distribution based on the set of QD optimised tasks. The paper’s experiments on GridNav, Alchemy, and Racing tasks show significant improvement in agent performance over existing baselines such as robust prioritised level-replay.

**Strengths:**

Focuses on high-level features more relevant to downstream tasks that the meta-RL agent should learn to adapt in, rather than focusing on simulator parameters as in prior unsupervised environment design (UED) work.

By shifting the focus in task distribution design for meta-RL training, this paper showed significant improvements on representative existing UED solutions, on a diverse set of evaluation tasks (GridNav, Alchemy, Racing). The evaluations are well-designed with sufficient ablations.

Makes a connection between QD and UED for meta-RL training, which is novel to the best of this reviewer’s knowledge.

The paper is very clearly written and illustrated. Relevant related works are discussed and contributions of the work are put into appropriate context. Content is well self-contained despite introducing new methods and tasks.

**Weaknesses:**

It is still necessary to hand-craft the high-level features, which needs expert knowledge (or at least quite high familiarity with downstream tasks) and can be heuristic.

Evaluation only used one meta-RL method (VariBAD). It would be useful to see performance of other meta-RL methods such as RL^2 on the DIVA task distribution.

**Questions:**

Does DIVA depend on any parameterizations at all? It seems like it might be possible (and very useful) to try DIVA on “weakly” parameterized tasks such as those generated by prompting an LLM or, say Genie [1].

[1] Genie: Generative Interactive Environments. Bruce el al. https://arxiv.org/abs/2402.15391

**Limitations:**

Limitations have been adequately addressed.

---

> ### Author Rebuttal · Authors · 2024-08-07
>
> We thank the reviewer for their thoughtful review our work and for appreciating its merits. We have addressed the noted weaknesses below, as well as their question about parameterizations.
>
> > [W1] It is still necessary to hand-craft the high-level features, which needs expert knowledge (or at least quite high familiarity with downstream tasks) and can be heuristic.
>
> We agree that this is a limitation of our work, which we discuss in Section 7 (Line 319). In our response to [W1] from Reviewer 7bxy, we have included discussion of an example work [1] where QD features are learned automatically; it is possible we can draw inspiration from works in this vein to further automatize DIVA.
>
> [1] Ding, L., Zhang, J., Clune, J., Spector, L., & Lehman, J. (2024). Quality Diversity through Human Feedback: Towards Open-Ended Diversity-Driven Optimization. In Forty-first International Conference on Machine Learning.
>
> > [W2] Evaluation only used one meta-RL method (VariBAD). It would be useful to see performance of other meta-RL methods such as RL^2 on the DIVA task distribution.
>
> It is true that all evaluations are conducted using VariBAD the base meta-RL algorithm. However, we did perform some preliminary evaluations with RL2—which is the same as VariBAD, except (1) instead of using a VAE with a latent space, we directly pass the RNN hidden state to the policy, and (2) we backpropagate the policy loss through the RNN (VariBAD only applies the VAE loss)—however, we did not see a drastic difference in the relative performance—between DIVA and its baselines—and because of the RNN backpropagate via the policy loss, training is significantly slower (as originally noted in [2]).
>
> Given these results, in absence of a performance differential, we decided on VariBAD, a state-of-the-art meta-RL algorithm. One reason we were drawn specifically to VariBAD is the harmony between the nature of task distributions, and VariBAD’s distributional nature. In DIVA’s case, task distributions are represented as a multivariate normal (some dimensions may be uniform) over a QD archive. We believe it may be possible for future works to exploit this connection; e.g. by using the location of a solution in the archive to ground the VariBAD latent space (a rough idea).
>
> [2] Zintgraf, L., Shiarlis, K., Igl, M., Schulze, S., Gal, Y., Hofmann, K., & Whiteson, S. (2019, September). VariBAD: A Very Good Method for Bayes-Adaptive Deep RL via Meta-Learning. In International Conference on Learning Representations.
>
> > [Q1] Does DIVA depend on any parameterizations at all? It seems like it might be possible (and very useful) to try DIVA on “weakly” parameterized tasks such as those generated by prompting an LLM or, say Genie [1].
>
> This is a great question, and certainly a direction worth exploring. DIVA in its current iteration can work with these weakly parameterized tasks, so long as (1) we have an environment generator that can accept this “weak” parameterization, and (2) we have access to some mutation method that can operate over this weak parameterization. For example, if the weak parameterization is language, we need some kind of language augmentation function (which prior literature has certainly explored). It may be even more effective to work with in the embedding space itself, since the augmentation function would be both simpler & smoother than language augmentation; in this case we would be essentially using QD to perform prefix/embedding tuning. We are excited to see these kinds of directions pursued in follow-up works.

---

> > ### Comment · Reviewer_kFbG · 2024-08-13
> > **Comment**
> >
> > Thank you for the detailed rebuttal. I maintain my original rating.

---

### Official Review · Reviewer_nCFY · 2024-07-12

**Soundness:** 2
**Presentation:** 3
**Contribution:** 2
**Rating:** 3
**Confidence:** 3

**Summary:**

This paper introduces DIVA, a technique for exploring the parameter space of parametrisable environments. The technique uses a variant of MAP-Elites to explore the environment parameter space, finding exemplar points spread across the parameter space, as measured with respect to some user provided features. The authors show that this generates a usefully diverse collection of environment instances by training a meta-learning agent on the generated levels and comparing it to a number of baselines.

**Strengths:**

On the whole the paper is well written and clear.

The authors choose a sensible selection of baseline algorithms to compare against, giving the reader an understanding of the strengths of their approach.

**Weaknesses:**

The primary weakness from my perspective is that domains that the experiments was conducted on are all very simple “toy” domains. In particular, these domains do not have all have intrinsically complex parameterisations, and for two of them the authors had to, in effect, obfuscate the parameter spaces so the algorithm had a challenge to work against. This leads one to worry that the authors’ results may not be representative of more realistic open-ended domains where the parameter space complexity may manifest in a different way.

Another weakness of the technique is the authors need to hand-select the features used by the algorithm, per domain. The authors mention that this could be automated, but the fact still stands that in this paper, quite extensive hand tuning and selection - as reported in the appendices - of the objectives was made to get their results. The impact of this technique is much more limited if feature sets need to be hand-tuned per domain, so without demonstrating that this is not the case, I think the authors do not demonstrate that this technique is likely to have wide impact.

I have a further query about the way that the VariBAD hyperparameters were tuned, below in the questions section.

**Questions:**

L36-37: Nit: “a” refers to the singular, “autocurricula” is plural.

L74: Nit: Add “are” between “algorithms” and “in”.

L129-131: Sentence doesn’t parse. Maybe remove “but”, or perhaps something got accidentally removed?

L159-161: I was confused by the experimental approach here. The downstream task distribution was first used to train the meta-learner’s hyperparameters, and then these hyperparameters were used to compare all of the approaches. But isn’t this essentially training on the test set? It feels like that the realistic setting would be one where the meta-learner does not know about the downstream tasks, and it would be important to demonstrate DIVA works under these conditions. I can appreciate that the authors might want to try to “factor” the behaviour of the meta-learner from the behaviour of DIVA, but it seems like a strong assumption that this is possible or useful.

L245: Nit: Figures 8 & 9 are referenced before figure 7.

L278: “Their approach”. Who’s approach?

L306: Define “HRI”.

L312: “DIVA’s” -> “DIVA”.

L312: “incorporating” -> “incorporate”.

**Limitations:**

The primary limitation is the one regarding hand-tuned feature sets, which I have expanded on above, in the weaknesses section.

---

> ### Author Rebuttal · Authors · 2024-08-07
>
> We appreciate the reviewer’s feedback and attention to detail. We have made all of the minor changes suggested under “Questions”, which will be present in the camera-ready draft. We address both major concerns below.
>
> **(1) On the domains being “toy”**
>
> > [W1] [...] domains [...] are all very simple [a] “toy” domains. In particular, these domains do not have all have [b] intrinsically complex parameterisations, and for two of them the authors had to, in effect, obfuscate the parameter spaces so the algorithm had a challenge to work against. [...] the authors’ results may [c] not be representative of more realistic open-ended domains where the parameter space complexity may manifest in a different way.
>
> **(1a)** For academic institutions with limited computational resources, this set of domains presents challenge enough to usefully benchmark methods, while keeping training runs cheap enough that many experimental trials can be run—enabling us to achieve statistically significant results.
>
> **(1b)** As discussed openly in Section 5, we recognize that the domains chosen require us to “obfuscate” the existing environment parameterization(s). We believe the reviewer may fundamentally misunderstand why adjusting the parameterization of these domains is necessary.
>
> Most meta-RL/RL domains are released with a parameterization/generator that can produce levels of meaningful diversity. Especially for domains with complex dynamics, designing these are not so straightforward. Because DIVA-like approaches are not widely studied at present, most domain designers do carefully hand-craft task distributions so that researchers will use their domains.
>
> As meta-RL approaches improve, and domains become more complex and open-ended, well-structured environment parameterizations will become increasingly expensive to implement by hand. Approaches like DIVA will enable learning on these domains, without requiring carefully hand-designing a structured parameterization and corresponding task generator. We believe that demonstrating the existence of DIVA-like methods that can handle ill-parameterized domains is a necessary step to inspire researchers to build more open-ended domains with unstructured parameterizations in the first place.
>
> **(1c)** The main property that makes a parameter space “unstructured” is the low probability of producing meaningfully diverse levels. This is the guiding principle used to design the challenging parameterizations for our evaluation domains—which, as we have explained above in response to (1b), was a necessity, since most (if not all) current academic domains are released with convenient parameterizations. While we agree that “the parameter space complexity may manifest in a different way” for more complex open-ended domains, what will remain is the overarching principle of meaningfully diverse levels having a low probability of being generated. Without detailing some other specific property of open-ended domains we might have overlooked, this weakness is a “whataboutism” that can be applied in similar form to any set of finite empirical findings.
>
> **(2) On hand-selected features**
>
> > [W2] […] The authors need to hand-select the features used by the algorithm, per domain. […] The impact of this technique is much more limited if feature sets need to be hand-tuned per domain, so without demonstrating that this is not the case, I think the authors do not demonstrate that this technique is likely to have wide impact.
>
> First, only a few features specified (a subset of those detailed for each domain in the Appendix) were used in experiments, and the final designs for each were guided more by intuition (e.g. looking at the distributions and covariance matrices) than any "extensive" experimental tuning. We also demonstrate with DIVA+ (Line 259) that the misspecification of features/objectives can be made up for by performing a small number of online evaluations like UED works.
>
> We believe that this criticism also overlooks the “extensive hand tuning” that is currently required to design parameterizations for complex domains. It is not that other approaches are able to sidestep the necessity of “hand-tun[ing]” feature sets per domain—it’s that this tuning is buried within the environment generation logic, which enabling a conveniently structured parameterization.
>
> The assumption of having access to some features of useful diversity is much weaker assumption than the assumption of having pre-existing generators that can produce levels across these same features of diversity. Consider—how does one design a structured parameterization/generator without first determining what the resulting level features should look like?
>
> **Questions**
>
> > [Q1] L159-161: [...] The downstream task distribution was first used to train the meta-learner’s hyperparameters, and then these hyperparameters were used to compare all of the approaches. But isn’t this essentially training on the test set? [...] I can appreciate that the authors might want to try to “factor” the behaviour of the meta-learner from the behaviour of DIVA, but it seems like a strong assumption that this is possible or useful.
>
> We recognize the ambiguity our wording may have caused here. The meta-learner is in fact “tuned” on the structured parameterizations. As can be seen in Appendix C.2, only a few VariBAD hyperparameters are adjusted per domain, which mostly pertain to pragmatic considerations such as network structure, and RAM/storage considerations. Because the parameterizations we consider produce poor training levels, we validate that the meta-RL component works by running the agent on the structured generator. This is indeed to “’factor’ the behavior of the meta-learner from the behavior of DIVA” and was a pragmatic decision so that we could more confidently study the effects of our method over the baselines. Once the meta-learner configuration was determined for each domain, it was held fixed for each method evaluated.

---

> > ### Author Response · Authors · 2024-08-08
> >
> > Important typo fix: under **Questions** [Q1]: "the meta-learning is in fact *not* tuned" (the *not* is missing in the current response, and we are unable to edit the response directly at this time).

---

> ### Author Response · Authors · 2024-08-07
> **Additional Rebuttal Content**
>
> More on **(1a)**: GridNav is a standard navigation task which serves as a useful didactic domain (and serves this purpose in VariBAD [1] as well), the Racing domain is a control domain used to benchmark numerous UED works [2, 3], and symbolic Alchemy is a chemistry-inspired meta-RL domain that requires complex trial-and-error reasoning over numerous episodes, which nicely complements the navigation and control tasks.
>
> More on **(2)**: We might also add that this is the first work to consider the combination of QD and meta-RL in this manner. In quality diversity (QD) literature, features are typically assumed to be pre-determined [4]. This pre-specification of features is indeed a limitation, but one that can be lifted in a similar fashion to QD works that learn features automatically [5]. Based on the contributions of our work, we decided it was appropriately out of scope to additionally consider the problem of learning features automatically.
>
> [1] Zintgraf, L., et al. (2019, September). VariBAD: A Very Good Method for Bayes-Adaptive Deep RL via Meta-Learning. In International Conference on Learning Representations.
>
> [2] Parker-Holder, J., et al. (2022, June). Evolving curricula with regret-based environment design. In International Conference on Machine Learning (pp. 17473-17498). PMLR.
>
> [3] Jiang, M., et al. (2021). Replay-guided adversarial environment design. Advances in Neural Information Processing Systems, 34, 1884-1897.
>
> [4] Pugh, Justin K. et al. (2016) “Quality Diversity: A New Frontier for Evolutionary Computation.” Frontiers in Robotics and AI 3:40.
>
> [5] Ding, L. et al. (2024). Quality Diversity through Human Feedback: Towards Open-Ended Diversity-Driven Optimization. ICML.

---

### Official Review · Reviewer_34nq · 2024-07-13

**Soundness:** 3
**Presentation:** 2
**Contribution:** 3
**Rating:** 6
**Confidence:** 3

**Summary:**

The paper identifies the limitation that hand-crafting a sufficiently diverse set of simulated training tasks to bridge any significant sim-to-real gap is labor-intensive. It then proposes DIVA, a new evolutionary approach for generating diverse training tasks in the absence of well-behaved simulator parameterizations. The paper demonstrates how DIVA outperforms proposed baselines such as ACCEL, PLR, and DR.

**Strengths:**

- The paper identifies a key limitation in current methods used to bridge the sim-to-real gap.
- The proposed approach is novel.
- There is a good spread of experimental results.

**Weaknesses:**

- I think the presentation of the paper can be improved. For example, Figure 1 could include more descriptions about what is happening, such as defining E_S(theta) as the structured environment simulator and E_U(theta) as the unstructured environment simulator.
- In line 128, how was the number 80% chosen? And why is it described as “roughly”? More justifications for the choice of these parameters should be included.

Minor things
- Typo in line 184, “the final y location is determine by”

**Questions:**

- Why does DIVA want to evolve a population of minimally diverse solutions from the original parameterization? Does DIVA assume that the original parameterizations are what humans care about? If so, doesn’t that mean we still need to ensure that the original parameterizations are handcrafted enough to be close to what humans care about?
- If the original parameterizations are chosen randomly or are not handcrafted to what humans might care about, how would DIVA perform?
- To what extent and when do we consider an environment simulator to have structured or unstructured parameterizations?
- In the GridNav experiment, why do we want to prevent the generation of diverse goals along the y-axis?
- Is there a hypothesis on why “the domain randomization over increasingly complex genotypes diminishes diversity in terms of goal location” (lines 187-188)?
- Line 188 claims that “DIVA, on the other hand, is able to capture this diversity.” However, doesn’t DIVA also decrease to the same level of percentage coverage as DR*? Why are there no confidence intervals for Figure 3a, like the others?

**Limitations:**

- Yes, the paper sufficiently covers the limitations.

---

> ### Author Rebuttal · Authors · 2024-08-07
>
> We thank the reviewer for their thoughtful review of our work, and for engaging us with curiosity to better understand the paper. Below we address the two main weaknesses noted by the reviewer, and clarify certain aspects of the work in response to the reviewer’s questions.
>
> > [W1] [...] The presentation of the paper can be improved. For example, Figure 1 could include more descriptions about what is happening [...]
>
> We appreciate this constructive feedback. We have updated Figure 1 (see Visual A in the 1-page supplemental), which now includes more descriptions as suggested by the reviewer.
>
> Additionally, we have taken other efforts to improve the presentation quality of the rest of the paper. These changes include adding algorithmic pseudocode to the Appendix (Visual E in the 1-page PDF) for the sake of clarity, a number of minor updates in wording for the sake of exposition clarity, switching out pixelated plots for raw vectorized PDFs for better image resolution, and some other minor aesthetic changes along these lines.
>
> We welcome any other concrete suggestions the reviewer may have for improving the presentation.
>
> > [W2] In line 128, how was the number 80% chosen? And why is it described as “roughly”?
>
> We thank the reviewer for pointing out this ambiguity. By “roughly” we were referring to the nature of samples versus population: we set this threshold using the 80th percentile of the samples, which encapsulates roughly 80% of the population. We understand this wording is unclear, and have therefore clarified what we mean precisely in the manuscript.
>
> > […] More justifications for the choice of these parameters should be included.
>
> 80% was chosen as an intuitive heuristic to balance the need for preserving the diversity contained within the DR samples used to initialize the archive (and the corresponding region that may be filled with useful mutations), and given a fixed archive size, maintaining as high a resolution as possible to enable the fastest propagation of solutions to and within the target region.
>
> We have run experiments to demonstrate that any moderate setting of this value prevents the failure modes of either extreme—see Visual D in the 1-page PDF (which will be included in the camera-ready, along with relevant discussion). In Alchemy (left) we see that the failure mode preventing the generation of target solutions is when we set this too low—-i.e. where we take only a small part of the tail of the DR samples. In Racing (right) we see the opposite problem; when set too high, it has significantly reduced the number of target samples. In both cases, the any moderate settings of the hyperparameter seems to do the job; it’s really about preventing either of the extremes. We chose 80% to be on the conservative side, but from the experiments, it looks like a setting of around 50% might even be better (from the Racing results).
>
> Additionally, we include robustness studies for other major hyperparameters in Visuals B, C, and F in the 1-page PDF (also for camera-ready). All attached figures correspond to Alchemy, but we can also provide similar ablations for Racing in the camera-ready draft. In Visual B we show robustness of final returns based on the mutation rate (Alchemy); in Visual C we see more QD updates is generally better, but our setting of 80k is not necessary to produce significant benefits over baselines; and in Visual F we see that while estimates for the normal parameters become less accurate with fewer sample features provided, DIVA is able to outperform baselines with as little as 5 downstream feature samples.
>
> > [W3] Typo in line 184, “the final y location is determine by”
>
> Fixed, thanks!
>
> Questions
>
> > [Q1] Why does DIVA want to evolve a population of minimally diverse solutions from the original parameterization? Does DIVA assume that the original parameterizations are what humans care about? If so, doesn’t that mean we still need to ensure that the original parameterizations are handcrafted enough to be close to what humans care about?
>
> We believe there is a key misunderstanding here between the function of DIVA and the experimental setup. We choose parameterizations in our evaluation that do not provide the dials—so to speak—to tune the features that humans care about (like the dials on $E_S(\theta)$ in Figure 1—-updated version in 1-page PDF response, Visual A). And these challenging parameterizations are what produce the “minimally diverse” solutions with which the archive is initialized. DIVA is designed to work with parameterizations like these that are explicitly not designed to produce diversity that humans care about. DIVA overcomes these difficult parameterizations by using QD to populate an archive of solutions that are diverse along certain features—which are hand-designed in this work, but may be learned automatically in future work (e.g. [1], a citation we’ve added thanks to the suggestion of Reviewer 7bxy).
>
> It is alternatively possible the reviewer might be asking if we need access to $E_S(\theta)$. DIVA does not need access to this generator/parameterization; all it needs is enough feature value samples from the downstream distribution to roughly estimate what our target distribution over the archive should be. It does not need the parameters that correspond to these scenarios. For example, if DIVA were tasked with generating house layouts for a house navigation task, it would need to know the target distribution for the number of rooms / the hallway width, etc. which could be determined from e.g. photos; but it would not need the parameters for constructing these sample houses in the simulator.
>
> In short, it is the features that must be designed/specified to reflect what humans care about, not the parameterization. Most works assume the parameterization is well-behaved. We challenge this assumption, and replace it with an assumption we believe is more realistic for open-ended environment simulators.

---

> ### Author Response · Authors · 2024-08-07
> **Additional Rebuttal Content (1)**
>
> > [Q2] If the original parameterizations are chosen randomly or are not handcrafted to what humans might care about, how would DIVA perform?
>
> We believe this question likely reflects the same misunderstanding noted above; we refer the reviewer to our [Q1] response, and welcome any follow-up questions the reviewer may have on this point.
>
> > [Q3] To what extent and when do we consider an environment simulator to have structured or unstructured parameterizations?
>
> We introduce the idea of structured/unstructured parameterizations in Figure 1, and elaborate specifically on the unstructured parameterizations DIVA can work with starting on Line 88. We define structured parameterizations as ones that enable random sampling to produce diverse levels of interest, whereas increasingly unstructured parameterizations produce these useful levels with diminishing probability. These terms are not absolute, but are instead relative, and represent two poles of a continuum. Open-ended domains, with more degrees of freedom and greater complexity in environment dynamics, must either be carefully parameterized and provided with a sophisticated generator that can navigate these complexities (i.e. a structured parameterization), or they can be more flexibility parameterized with a simpler generator (an unstructured parameterization), where meaningful diversity is possible, but not guaranteed with high probability. DIVA can make use these more unstructured parameterizations, and some knowledge of some knowledge of what levels should look like, in order to produce an abundance of training levels that are diverse in these ways. Importantly, this saves the prohibitive effort of having to carefully craft structured parameterizations / generators for complex open-ended domains.
>
> > [Q4] In the GridNav experiment, why do we want to prevent the generation of diverse goals along the y-axis?
>
> We believe this may pertain to the same misunderstanding addressed in [Q1], so we refer the reviewer to that response, but we also address this question more extensively below.
>
> The object of DIVA and the baselines are to produce meaningful diversity in the generated levels, and for the GridNav domain, this means producing goal diversity along both the x and y axis.
>
> In our GridNav experimental setup, we inject complexity into the parameterization that makes it difficult to generate goal diversity along the y-axis. We design the parameterization in such a way that we can vary the complexity (unstructuredness) of the parameterization. As we increase the complexity, diversity along the y-axis becomes less and less likely. Our results in GridNav show that DIVA is best able to capture this diversity by (1) evolving levels with the expressed purpose of discovering and preserving levels of meaningful diversity (via the QD archive), and (2) avoiding agent evals like ACCEL/PLR, which are expensive, and doesn’t guarantee all diversity discovered is preserved.
>
> > [Q5] Is there a hypothesis on why “the domain randomization over increasingly complex genotypes diminishes diversity in terms of goal location” (lines 187-188)?
>
> This is by design (described in Line 178) rather than some emergent property requiring hypothesis, and is likely relevant to the misunderstanding addressed in [Q4]. Increasingly complex genotypes diminish diversity in terms of goal location by our design of the genotype; and we design the general genotype scheme in such a way that we can conveniently vary the complexity—in this case, the number of genes that must coordinate to produce diversity along the y-axis. The effect of this design is that diversity along the y-axis becomes less and less likely (with random genotypes) as the complexity increases, because more genes need to coordinate.

---

> ### Author Response · Authors · 2024-08-07
> **Additional Rebuttal Content (2)**
>
> > [Q6] Line 188 claims that “DIVA, on the other hand, is able to capture this diversity.” However, doesn’t DIVA also decrease to the same level of percentage coverage as DR*? Why are there no confidence intervals for Figure 3a, like the others?
>
> Figure 3a shows the ability of DR, DR*, and DIVA to “capture” diversity in generated levels with parameterizations of varying complexity. DR* reflects the upper bound of what a method like PLR can achieve with DR levels—it is the maximum diversity see by DR (i.e. the total number of unique levels seen over the entire generation process), whereas DR is just the final archive produced. DR has no memory mechanism to preserve diversity, whereas a method like PLR, with its priority buffer, can latch on to the diversity contained in DR* (but still may fail to preserve all discovered diversity).
>
> The reviewer is correct that DIVA is shown to capture just as much as DR* at the most complex parameterization we tested. However, given the nature of the experimental design, this trailing off is destined to happen with a high enough complexity; it becomes statistically unlikely any algorithm to produce meaningful diversity. It was a purposeful decision to demonstrate that DIVA tails off in its ability to handle these highly challenging parameterizations. If we increased the number of update steps allowed for DIVA, or inserted a more sophisticated emitter algorithm (instead of MAP-Elites), DIVA would be able to benefit from this far more than DR* (the same trend would extend, but fall off just the same at a high enough complexity).
>
> The takeaway from this plot is that QD is able to uncover and preserve diversity in the archive as the complexity increases better than DR (or our invented oracle DR*). This is also why we have not included confidence intervals—-the point is to show the trend; each point is just a single run of QD updates. Given the cleanness of the trend, we found it unnecessary to run more seeds for this plot, but we can do this for the camera-ready draft. Either way, we have clarified this point in the paper. Thanks for your feedback!

---

> > ### Comment · Reviewer_34nq · 2024-08-08
> >
> > Thanks to the authors for their detailed response and the work put into the new experiments. The new experiments do address many of the concerns I had. For the presentation of the paper, I hope the authors will put the explanations for many of the design choices (like what the authors wrote for rebuttals above) into the revised manuscript. I think that will significantly improve the paper's presentation. As such, I have increased my score.

---

> > > ### Author Response · Authors · 2024-08-08
> > >
> > > We appreciate the reviewer for engaging with our rebuttal and for helping strengthen our paper. We will follow through with the reviewer's feedback to include the explanations for the design choices (including the content contained in our rebuttal) in the camera-ready manuscript. This will be added along with the ablations and other updates noted, which have already been applied to our working manuscript draft. We thank the reviewer for updating their score, and are more than happy to address any remaining concerns.

---

### Official Review · Reviewer_7bxy · 2024-07-13

**Soundness:** 3
**Presentation:** 2
**Contribution:** 3
**Rating:** 7
**Confidence:** 4

**Summary:**

This paper describes an approach for learning a QD-archive to be used as a proxy for samples of test environments, and shows that this results in improved performance in producing a set of meta-learning tasks over DR and UED baselines.

**Strengths:**

The introduction of QD approaches into UED algorithms is a promising algorithm for improvement. The empirical results appear convincing, and the approach is quite natural.

**Weaknesses:**

The comparison between DIVA and PLR/ACCEL is a bit of an apples-to-oranges comparison. Regret-based UED methods like PLR and ACCEL are meant to be making decisions under ignorance, where there is no information known about the target distribution. There are other UED approaches designed for the case where there is some information that is known, and thus it is a decision under risk. Specifically, it would be better to compare against SAMPLR, CLUTER, or DRED. Since SAMPLR requires simulator access, the fairest comparison would be against CLUTER or DRED.

[SAMPLR] Jiang, Minqi, et al. "Grounding aleatoric uncertainty for unsupervised environment design." _Advances in Neural Information Processing Systems_ 35 (2022): 32868-32881.

[CLUTER]Azad, Abdus Salam, et al. "Clutr: Curriculum learning via unsupervised task representation learning." _International Conference on Machine Learning_. PMLR, 2023.

[DRED] Garcin, Samuel, et al. "DRED: Zero-Shot Transfer in Reinforcement Learning via Data-Regularised Environment Design." _Forty-first International Conference on Machine Learning_. 2024.

That being said, the current results look to have more significant results than these other works (though it is hard to tell across domains), and this approach has not be tried in meta-learning before. It is important to note that UED has been used in meta-learning before, for instance in AdA.

[AdA] Team, Adaptive Agent, et al. "Human-timescale adaptation in an open-ended task space." _arXiv preprint arXiv:2301.07608_ (2023).

It occurs to me that DIVA + UED as discussed starting at line 259 could be used as an algorithm for decisions under ignorance, and would possibly be quite a good algorithm. It may be worth running the approach from 259, maybe without any test time data, on the traditional maze, F1, and bipedal walker environments and transfer tasks to check if it consistently outperforms existing methods.

It seems like the assumption about the probability of generating a useful level from the simulator underlying DIVA is similar to the assumption necessary for ACCEL or PLR. Both of these approaches should work if there is a ~0.01 % chance of generating a useful level. The boot-up time would just be a bit slower to get the initial buffer of levels.

The limitation of online agents evaluations is a real bottleneck for the UED approaches which DIVA avoids, but it is a bit of an odd comparison as DIVA generates the model once offline, and thus isn't aiming to be adaptive towards current agent performance. It seems like DIVA looses the benefits of adaptivity by completely removing online agent evaluations. I would be interested if a DIVA-like approach with limited online evaluations could keep the best of both.


In Figure 5d, "unique genotypes is a quite bad metric for diversity, as completely random levels would score quite well even though most random levels are quite qualitatively to each other.

### Clarity

In the abstract it is not clear to me what "well-behaved simulator parameterisations" means, what "unscalable flexibility" refers to, or what "ill parameterised simulators" means.

It would be good to have a citation on line 46 for how one could learn the axes of QD.

It's not immediately clear what "genotype" means, and it seems to be used in different ways on line 82 and 88. In the first case it may be more conventional to call it the "level generator" and in the second case it could be more conventional to call it the "level parameters".

**Questions:**

How would this method compare to CLUTR or DRED?

**Limitations:**

See Weaknesses.

---

> ### Author Rebuttal · Authors · 2024-08-07
>
> We thank the reviewer for taking the time to write such a thorough review, and for not only appreciating the merits of this work, but for identifying points of weakness to strengthen our manuscript. We have responded to each of the weaknesses listed and questions posed below, and look forward to further discussion.
>
> > [W1] The comparison between DIVA and PLR/ACCEL is a bit of an apples-to-oranges comparison. [...] the fairest comparison would be against CLUTER or DRED.
> >
> > [Q1] How would this method compare to CLUTR or DRED?
>
> We have added each of the suggested approaches (SAMPLR, CLUTR, and DRED) to the related works section, which will appear in our camera-ready draft. Our assessment of each of the individual works—see conclusions below, and **full assessments attached as "official comments"**—is that none are as appropriate a baseline choice as ACCEL, despite some similarities (especially DRED) to our problem setting.
>
> **SAMPLR**  We agree with the reviewer’s assessment that SAMPLR is less relevant for direct comparison to DIVA since SAMPLR requires access to the simulator itself (and the direct downstream sampling distribution, which we do not assume access to either). However, we believe SAMPLR’s relevance to the meta-RL setting (meta-RL potentially suffers even more from this bias) warrants mention in the related work section---an addition which will appear in the camera-ready draft.
>
> **CLUTR** By training a VAE on levels sampled from the generator distribution, which lack greatly in diversity in DIVA’s setting, CLUTR would fail to learn a generator that produces phenotypically diverse levels. ACCEL, by performing mutations to existing levels, is able evolve its levels towards greater diversity over time, making it a stronger baseline for our setting. Due to its relevance to future work, we have cited and added discussion of this work to the manuscript, which will appear in the camera-ready draft.
>
> **DRED**  The biggest difference between DRED and DIVA’s settings is that DIVA does not assume access to the underlying level parameters, but rather only the feature values---and these alone are used to define the archive. This difference makes DRED suffer from the same issue as CLUTR when faced with the setting that DIVA operates in: if the parameter space cannot produce diverse levels for the VAE training, then the resulting model will produce levels just as lacking in diversity. Resampling a la PLR will have a similar effect as our PLR baseline. Thus, as was mentioned of CLUTR above, ACCEL remains a stronger baseline than DRED for this reason. That being said, DRED’s close resemblance to our setting and possible applicability to future work (like CLUTR), we have cited and added discussion to our manuscript, which will appear in the camera-ready draft.
>
> > [W2] It occurs to me that DIVA + UED [...] could be used as an algorithm for decisions under ignorance, and would possibly be quite a good algorithm. It may be worth running [...] on the traditional maze, F1, and bipedal walker environments and transfer tasks to check if it consistently outperforms existing methods.
> >
> > [...]
> >
> > The limitation of online agents evaluations is a real bottleneck for the UED approaches which DIVA avoids, but it is a bit of an odd comparison as DIVA generates the model once offline, and thus isn't aiming to be adaptive towards current agent performance. It seems like DIVA looses the benefits of adaptivity by completely removing online agent evaluations. I would be interested if a DIVA-like approach with limited online evaluations could keep the best of both.
>
> We would like to note that the DIVA+ algorithm (which demonstrating learning benefits when the archive is “misspecified”) is indeed a “DIVA-like approach with limited online evaluations”, so the reviewer is correct in identifying the promise of such a combination. And we agree that this kind of approach warrants more exploration; we include these preliminary results simply to showcase the promise of an approach of this nature as a potential future direction, and leave it to future work to both (1) provide a more principled integration of UED/QD (our DIVA+ algorithm is just one such combination), and to (2) produce more extensive results with such an approach on relevant domains, such as the ones the reviewer has listed.
>
> > [W3] It seems like the assumption about the probability of generating a useful level from the simulator underlying DIVA is similar to the assumption necessary for ACCEL or PLR. Both of these approaches should work if there is a ~0.01 % chance of generating a useful level. The boot-up time would just be a bit slower to get the initial buffer of levels.
>
> While the reviewer is correct in highlighting that this general assumption is shared between DIVA and ACCEL (and PLR), we demonstrate that DIVA is able to work with generators that produce smaller percentages of useful levels. In practice, the boot-up time for ACCEL (and PLR) is more than a bit slower. Because ACCEL must simulate the agent on each environment, and DIVA only requires rendering the first timestep, then ACCEL is $|\tau| \times H$ times slower if we wish to perform just as many evolutionary updates as DIVA (in practice rendering the first timestep to compute features may be slower depending on the environment, but the more general point still holds). Because DIVA can produce far more evolutionary updates in the same amount of time by avoiding simulating the agent on every level (and DIVA+ avoids this too by only simulating levels that constitute the final archive, after all the initial QD updates), DIVA can work more effectively with environments where generating a useful level is even rarer. This is a point we tried to get across in Section 3 (Problem setting), but for the camera ready we will also include the reasoning we’ve provided above—we thank the reviewer for prompting us to consider this point more closely in our discussion!

---

> ### Author Response · Authors · 2024-08-07
> **Additional Rebuttal Content**
>
> > [W4] In Figure 5d, “unique genotypes[“] is a quite bad metric for diversity, as completely random levels would score quite well even though most random levels are quite qualitatively to each other.
>
> We agree with the reviewer that unique genotypes is indeed a bad metric for diversity. We include this plot not to highlight diversity, but to make the same observation the reviewer notes (see Line 207)—-that despite DR (for example) producing more unique genotypes than DIVA, DIVA is able to produce greater meaningful diversity, which leads to better performance. We also note that in Line 208 we mistakenly referenced “Figure 5” (in general) instead of “Figure 5d”, and we have made this correction.
>
> > [Q2] In the abstract it is not clear to me what "well-behaved simulator parameterisations" means, what "unscalable flexibility" refers to, or what "ill parameterised simulators" means.
>
> We agree with the reviewer that the abstract should be self-contained, and that terminology should either be widely understood or defined in the abstract itself. We will modify these phrases in the camera-ready draft to be clear and unambiguous. We thank the reviewer for bringing our attention to these instances.
>
> > [Q3] It would be good to have a citation on line 46 for how one could learn the axes of QD.
>
> Great suggestion; we have added one such example [1] to our manuscript to be present in the camera-ready draft.
>
> [1] Ding, L., Zhang, J., Clune, J., Spector, L., & Lehman, J. (2024). Quality Diversity through Human Feedback: Towards Open-Ended Diversity-Driven Optimization. In Forty-first International Conference on Machine Learning.
>
> > [Q4] It's not immediately clear what "genotype" means, and it seems to be used in different ways on line 82 and 88. In the first case it may be more conventional to call it the "level generator" and in the second case it could be more conventional to call it the "level parameters".
>
> The reviewer is correct that the first use of genotype in Line 82 is misleading; we will update to “level parameterization” as this is what we mean. In Line 88 we will use “level parameters” in as the reviewer suggests for clarity, but we will still introduce and use “genotype” as a shorthand/alternative for “level parameters”, in order to respect the connection to evolutionary approaches.

---

> ### Author Response · Authors · 2024-08-07
> **SAMPLR, CLUTR, and DRED Full Assessments**
>
> > [SAMPLR] Jiang, Minqi, et al. "Grounding aleatoric uncertainty for unsupervised environment design." Advances in Neural Information Processing Systems 35 (2022): 32868-32881.
>
> **SAMPLR** attempts to correct the “curriculum-induced covariate shift” (CICS)—with respect to the downstream task distribution—that results from learning an adaptive curriculum. SAMPLR mitigates this bias over aleatoric parameters while preserving the benefits of utilizing a curriculum (versus sampling directly from the downstream task parameters).
>
> Assessment: We agree with the reviewer’s assessment that this approach is less relevant for direct comparison to DIVA since SAMPLR requires access to the simulator itself (and the direct downstream sampling distribution, which we do not assume access to either). However, we believe SAMPLR’s relevance to the meta-RL setting (meta-RL potentially suffers even more from this bias) warrants mention in the related work section---an addition which will appear in the camera-ready draft.
>
> > [CLUTR] Azad, Abdus Salam, et al. "Clutr: Curriculum learning via unsupervised task representation learning." International Conference on Machine Learning. PMLR, 2023.
>
> The authors of **CLUTR** argue that many UED methods are burdened by the difficulty of simultaneously learning the task manifold (implicitly through RL, for PAIRED-based methods) and the curriculum over these tasks. CLUTR disentangles the task representation learning from curriculum learning by first pretraining a task manifold with a VAE in an unsupervised manner, and then learning a curriculum over this (fixed) task manifold via maximizing regret.
>
> Assessment: By training a VAE on levels sampled from the generator distribution, which lack greatly in diversity in DIVA’s setting, CLUTR would fail to learn a generator that produces phenotypically diverse levels. ACCEL, by performing mutations to existing levels, is able evolve its levels towards greater diversity over time, making it a stronger baseline for our setting. CLUTR may, however, be a useful future addition to a DIVA-like approach, e.g. by converting the archive to a learned manifold via a VAE. This could be potentially useful for storage considerations (i.e. if our archive is too large), or to extrapolate beyond levels that exist in the archive. Performing curriculum learning over this distilled archive would look something like our DIVA+ algorithm, which combines the benefits of DIVA (diversity) with UED (curriculum) approaches, but would have these storage/extrapolatory benefits. Due to its relevance to future work, we have cited and added discussion of this work to the manuscript, which will appear in the camera-ready draft.
>
> > [DRED] Garcin, Samuel, et al. "DRED: Zero-Shot Transfer in Reinforcement Learning via Data-Regularised Environment Design." Forty-first International Conference on Machine Learning. 2024.
>
> The authors introduce data-regularized environment design (**DRED**), which combines adaptive sampling (like UED) with a level generator that approximates p(x). DRED assumes access not directly to p(x), but to a set of level parameters X with which a VAE can be trained. Once the VAE is trained, PLR is used to produce scores for sampling (other details omitted here for simplicity).
>
> Assessment: The biggest difference between DRED and DIVA’s settings is that DIVA does not assume access to the underlying level parameters, but rather only the feature values---and these alone are used to define the archive. The QD search is then tasked with producing samples that match the feature distribution (in our work, for both simplicity and sample efficiency, making assumes of either independent Gaussian/Uniform distributions). DIVA’s archive is initialized with random parameters from the parameter space; it never sees the parameters corresponding to the feature values. This difference makes DRED suffer from the same issue as CLUTR when faced with the setting that DIVA operates in: if the parameter space cannot produce diverse levels for the VAE training, then the resulting model will produce levels just as lacking in diversity. Resampling a la PLR will have a similar effect as our PLR baseline. Thus, as was mentioned of CLUTR above, ACCEL remains a stronger baseline than DRED for this reason. That being said, DRED’s close resemblance to our setting and possible applicability to future work (like CLUTR), we have cited and added discussion to our manuscript, which will appear in the camera-ready draft. We would like to note, however, that DRED’s absense in our original discussion was due to its very recent publication (uploaded to ArXiv in February 2024; presented at ICML in July 2024).

---

> > ### Comment · Reviewer_7bxy · 2024-08-11
> > **Response to Rebuttal by Authors**
> >
> > Thank you for the detailed response, the proposed changes do clarify the paper significantly. Unfortunately I don't see room to increase my score past "high impact on at least one sub-area", but I appreciate the work and do believe this paper should be accepted.

---

### Author Rebuttal · Authors · 2024-08-07

We thank the reviewers for the time they have taken to engage constructively with our work. Reviewer _kFbG_ appreciates the “novel [...] connection between QD and UED for meta-RL training” we have developed, along with our work’s explicit “focus on [relevant] high-level features”—in contrast to the implicit assumptions of diversity along these features in prior works—and the resulting “significant improvements [on] well-designed” evaluations. Reviewer _7bxy_ finds the approach “natural”, and the results “convincing”. Reviewer _34nq_ believes the paper “identifies a key limitation” in existing literature and contains a “good spread of experimental results”. Reviewer _nCFY_ finds the paper “well written and clear”, and that a “sensible selection of baseline algorithms” are used for evaluation.

There are four main weaknesses in total that appear to be sticking points for Reviewers _34nq_ and _nCFY_, specifically. We have addressed each of these points at length in the individual responses, and summarize the holistic discussion of each point here.

**(1) Specifying features.**  A commonly noted limitation among the reviewers, and one we address ourselves in Section 7 of the manuscript, is that the “axes of diversity must be specified” (Line 319). Reviewer _kFbG_ views DIVA’s explicit “focus on [relevant] high-level features” as a strength of our work, an aspect which contributes to our paper identifying a “key limitation” (Reviewer _34nq_) in existing literature. As we point out in a number of individual responses, while QD literature often assumes access to these high level features, some works (e.g. [1]) are able to determine these automatically. We view this as an interesting avenue for future work, but outside the scope of this paper, which, as previously noted is already a “novel [...] connection between QD and UED for meta-RL training” (Reviewer _kFbG_). For Reviewer _nCFY_ alone it is a major sticking point from which they conclude (in addition to other human involvement in feature/objective selection) that we have “not demonstrate[d] that this technique is likely to have wide impact”. In response, we remind the reviewer that, despite explicitly making this new assumption ourselves, we are removing a much stronger assumption implicit in prior works:

> The assumption of having access to some features of useful diversity is much weaker assumption than the assumption of having pre-existing generators that can produce levels across these same features of diversity. Consider—how does one design a structured parameterization/generator without first determining what the resulting level features should look like?

See our full response to Reviewer _nCFY_ for more discussion on this point.

**(2) Presentation.**  Reviewer _34nq_ expresses concern over the presentation of the paper as their main weakness. Specifically, the example they give for where clarity can be improved is Figure 1. We update Figure 1 along the lines of the reviewer’s suggestions (see **Visual A in the attached PDF**), add algorithmic pseudocode to the Appendix (see **Visual E in the attached PDF**) to further elucidate our approach, and make some other minor formatting changes to improve the presentation for the camera-ready draft. It is worthwhile noting that beyond typos and updates to specific cases of ambiguous wording, no other major presentation concerns were noted by Reviewer _34nq_ or the other reviewers. Reviewer _nCFY_ says “on the whole the paper is well written and clear”, and Reviewer _kFbG_ finds the paper “very clearly written and illustrated”.

**(3) Hyperparameters.**  Reviewer _34nq_ expresses confusion over one of our hyperparameter choices (setting an archive bound based on 80% of samples), and expresses more generally that these decisions are not better justified in the submitted draft. We clarify our wording and justify our decision for the 80% bound hyperparameter, and provide a robustness study on both Alchemy and Racing domains to verify the intuition that any moderate setting of this hyperparameter avoids failure modes (see **Visual D in the attached PDF**). We additionally provide studies for other hyperparameter, demonstrating DIVA’s robustness to their settings (see **Visuals B, C, and F**). For more details, see our response to Reviewer _34nq_ [W2].

**(4) Evaluation domains.**  Reviewer _nCFY_’s main sticking point (unique among the reviewers; who otherwise find the evaluation sufficient and convincing) is that the domains are “toy”, specifically because we “obfuscate” the parameter spaces to produce challenging parameterizations. We first dispute the notion that the domains themselves are overly toy, both in terms of the unique challenges they pose for methods in our problem setting, and according to standards of academic research in meta-RL and UED. We then explain why it was necessary to reparameterize the tasks—namely, that most domains come pre-equipped with a parameterization that can generate levels of meaningful diversity. For open-ended domains of the future, with more complex dynamics, well-structured environment parameterizations will not be as feasible to implement by hand. Approaches like DIVA will enable learning on these domains, without requiring carefully hand-designing a structured parameterization or task generator. And crucially, we believe that demonstrating the existence of DIVA-like methods that can handle ill-parameterized domains is a necessary step to inspire researchers to build more open-ended domains with unstructured parameterizations in the first place.

[1] Ding, L., Zhang, J., Clune, J., Spector, L., & Lehman, J. (2024). Quality Diversity through Human Feedback: Towards Open-Ended Diversity-Driven Optimization. In Forty-first International Conference on Machine Learning.

---

> ### Comment · Reviewer_nCFY · 2024-08-09
>
> In response to the authors' detailed responses to my and other reviews, I have two comments:
>
> On the question of evaluation domains and their parametrisation. The authors note that the reason they deliberately re-parametrise their domains is because most existing evaluation domains are built with meaningful parametrisations by design. I think this point was already understood. My original comment was not to question why this was done, but rather to highlight that this introduces a critical weakness to the paper's argument: that DIVA is likely to be a useful technique for domains where a meaningful parametrisation is naturally not available.
>
> I don't think this criticism is "whataboutism" as the authors' poetically put it. It is well known - perhaps an "open secret" - that while genetic techniques are in principle very general, their efficacy in real-world scenarios often depends critically on the structure of the genotype-phenotype mapping. Indeed, if this map has no exploitable structure then genetic techniques generally reduce to random search. My fear is that by introducing their own simple "scrambling" of the genotype-phenotype mapping, that they have inadvertently introduced structure in this map which results in compelling performance from their algorithm. But it is certainly not obvious whether the required exploitable structures exist in more complex open-ended environments.
>
> To put it succinctly, the authors' main claim is that "Our empirical results demonstrate DIVA's unique ability to leverage ill-parameterized simulators to ..." and I'm not sure this is well supported, as the environments the authors have used may well not be "ill-parametrized" in a relevant way, with respect to more interesting open-ended environments.
>
> On specifying features. The authors' comment that while they appreciate hand-crafting diversity measuring features is a limitation, it is significantly less limiting than having to build a diverse environment generator. This comment is well taken. However, I think my objection is more that the authors make quite a big claim, that "These findings highlight the potential of approaches like DIVA to enable training in complex open-ended domains, and to produce more robust and adaptable agents" and I don't think that claim is well supported, given that they have *introduced* a new scalability bottleneck. This is less a comment on the research, and more a comment on how it is pitched. I would recommend softening the claim a little so that it is more consistent with what the authors have achieved.
>
> Overall, the authors comments don't significantly change my opinion of the paper's suitability for NeurIPS.
>
> [Edited for formatting only.]

---

> > ### Author Response · Authors · 2024-08-11
> >
> > We appreciate the reviewer’s continued engagement with our work, and for further clarifying their criticisms. We understand the reviewer’s desire to ensure the work’s claims are properly qualified and its significance is not overstated. However, we believe certain characterizations disputed by the reviewer are in fact accurate, which we address in detail below. We are happy to respond to any further questions or concerns the reviewer may have on these points or any others.
> >
> > ## (1) On the structure of open-ended parameterizations
> >
> > We would like to first address the reviewer’s “fear” that we might have “inadvertently introduced structure [that results] in compelling performance” for our algorithm specifically. **We can scrutinize the concrete facts of our experimental setup *directly* to determine if this fear is grounded in truth.** Let us remind ourselves of the parameterizations for both the Alchemy and Racing domains, which contain two very different types of open-ended structures.
> >
> > For the **Alchemy** domain, we effectively introduce a deeper layer of chemistry to the environment, whereby each latent stone dimension is represented by $k$ bits (of information) instead of just one. This increased resolution makes the environment more “open-ended”; greater diversity is created in the genotype space, but now it is less likely that random genotypes will produce the kind of diversity we care about. Thus, it is also a less “structured” parameterization, according to our definition; for stones with diverse phenotypes to be generated, the values of these bits must coordinate, which is made statistically less likely the higher $k$ is ($k = 8$ in our experiments; where the degenerate case is $k=1$).
> >
> > **DIVA is able to successfully coordinate these bits to produce diverse phenotypes *not* by exploiting the genotypical structure in some unique way that favors DIVA specifically**—-this can be seen in the mere fact that ACCEL uses the same mutation function as DIVA. Instead, *DIVA succeeds because (1) it is able to perform significantly more mutations and evaluations than ACCEL, by virtue of avoiding agent rollouts on every mutated level, and (2) by employing a QD archive, it captures all diversity encountered*, whereas approaches like ACCEL and PLR only store levels where learning potential is detected, meaning they will discard or ignore levels in the buffer where agent rollouts do not register learning potential.
> >
> > In **Racing**, a different kind of open-endedness challenge is introduced into the level parameter space. We define a small region of the parameter space that can maximally control the diversity of the racetracks. There is no structure or information about this region that is leaked to DIVA; the mutation function is simply fighting against the statistical improbability of landing in this region (which DIVA's QD archive is able to effectively capture). In Racing, as in Alchemy, **ACCEL and DIVA share the same mutation function**.
> >
> > To further justify these design choices, **consider what will naturally occur as simulated robotics domains become more “open-ended”**---we might make a robotic manipulation task more open-ended by enabling more granular objects to be represented and available for manipulation. But if we loosen the parameterization of the environment to allow a greater number of objects to be represented, it will be possible to specify all kinds of objects that either wouldn’t naturally occur in the real world, contain structural impossibilities, or otherwise would not be of much use to learn how to manipulate. Only a subset of the parameter space now will correspond to meaningfully diverse objects—-just like we have achieved for Alchemy and Racing. For Alchemy, we designed this diverse subset of configurations to correspond to a specific kind of improbable bit coordination, and in Racing, we designed this subset to correspond to a small region of the parameter space—-and in neither case does this structure specifically favor DIVA in any way.
> >
> > And to be clear why we have not obtained results on a robotics environment of the kind imagined above—-i.e. one with a “naturally” challenging open-ended parameterization: In short, this would have made for a prohibitively expensive initial exploration into this problem setting---especially given the limited resources of an academic institution, and the fact that meta-RL methods still find existing robotics domains challenging. We therefore stand by our decision to take domains that have proven useful for prior works UED and meta-RL research and bestow upon them the most important properties of domains with open-ended parameterizations.
> >
> > *[continued below]*

---

> > > ### Author Response · Authors · 2024-08-11
> > >
> > > *[continued from above]*
> > >
> > > **We would appreciate if the reviewer were able to provide some specific property of open-ended domains with unstructured parameterizations that we have not captured** with the principle of rarity in generating meaningfully diverse solutions. We do not claim that our evaluation covers every possible way in which open-ended domain parameterizations may express themselves, but this is not the burden of proof we place on methods for demonstrating their promise on a certain problem setting. Nor do we claim that DIVA can overcome shear statistical impossibilities (in practical terms) that some unstructured parameterizations may present. But based on our confidence in the experimental setup, and the convincingness of the results, we stand by the claim the reviewer has cited, that “our empirical results demonstrate DIVA’s unique ability to leverage ill-parameterized simulators [...]”.
> > >
> > > A brief note on **genetic techniques and QD**. It is possible that different QD algorithms will benefit more or less from whatever structure the genotype-phenotype mapping contains. For instance, in Alchemy, we use MAP-Elites over the discrete genotype, which does not exploit the structure at all. The random permutations uncover levels with new features periodically, and these levels, which have latched onto new diversity, are used as a basis for further mutations. DIVA can benefit from this in the ways discussed above, but it does not latch onto the structure in any way. For Racing, both DIVA and Alchemy use CMA-ES, a more sophisticated QD approach that works well with continuous action spaces—in this case, the covariance matrix guiding the mutations is adapted to more efficiently uncover the parameter space. As we mention in Section 7, it may be possible to even further inform training over complex action spaces by performing a pretraining step (or iterative updates)with some latent model, and then exploring the parameter space with a QD approach that can take advantage of gradients.
> > >
> > > ## (2) Specifying features
> > >
> > > We appreciate the receptivity of the reviewer on the point about hand-crafted generators, and their clarification on the nature of their objection surrounding features. We are not quite sure what the reviewer is referring to by the “new scalability bottleneck”, but we believe this likely refers to the assumption we make about the availability of specified features. If this is the case, we are unsure how the claim oversells our approach, given our transparency surrounding this limitation in the manuscript—which as previously mentioned, is a limitation shared by most QD works. Would the reviewer like to see, for instance, a more precise mention of this fact in the abstract, specifically? We also wonder, more fundamentally, how this qualifies as a “new scalability bottleneck”, given our agreement that designing diverse generators presents a larger scalability bottleneck than DIVA’s requirement of specifying features. It is because DIVA does away with this more restrictive bottleneck that we find it accurate to say that our “findings highlight the potential of approaches like DIVA to enable training in complex open-ended domains [...]”. This claim does not imply that all complex open-ended domains are now in reach; instead, we are trying to get across that DIVA has demonstrated the *potential* of these kinds of approaches for bringing open-ended domains *more* within our collective reach.

---

### Decision · Program_Chairs · 2024-09-25

**Decision:**

Accept (poster)

**Comment:**

The paper proposes a method for generating actually diverse training tasks.

Reviewers recommendations were Accept, Weak Accept, Reject, Strong Accept. There was some criticism of the fact that the proposed approach still requires some level of hand-design, and the simplicity of the experiments.

Overall I am willing to accept due to the interest of the problem and the novelty of the method.

I would recommend the authors re-read the paper and fix the  typos. E.g. line 130 "but...",  line 194: "the each stone".